# Selective on-surface covalent coupling based on metal-organic coordination template

Shuaipeng Xing[1], Zhe Zhang[2], Xiyu Fei[3], Wei Zhao[4], Ran Zhang[5], Tao Lin[6], Danli Zhao[3], Huanxin Ju[7], Hu Xu[2], Jian Fan[3], Junfa Zhu [7], Yu-qiang Ma[1,8] & Ziliang Shi [1]

Control over on-surface reaction pathways is crucial but challenging for the precise construction of conjugated nanostructures at the atomic level. Herein we demonstrate a selective on-surface covalent coupling reaction that is templated by metal-organic coordinative bonding, and achieve a porous nitrogen-doped carbon nanoribbon structure. In contrast to the inhomogeneous polymorphic structures resulting from the debrominated aryl-aryl coupling reaction on Au(111), the incorporation of an Fe-terpyridine (tpy) coordination motif into the on-surface reaction controls the molecular conformation, guides the reaction pathway, and finally yields pure organic sexipyridine-*p*-phenylene nanoribbons. Emergent molecular conformers and reaction products in the reaction pathways are revealed by scanning tunneling microscopy, density functional theory calculations and X-ray photoelectron spectroscopy, demonstrating the template effect of Fe-tpy coordination on the on-surface covalent coupling. Our approach opens an avenue for the rational design and synthesis of functional conjugated nanomaterials with atomic precision.

[1] Center for Soft Condensed Matter Physics and Interdisciplinary Research and School of Physical Science and Technology, Soochow University, 215006 Suzhou, China. [2] Department of Physics, Southern University of Science and Technology, 518055 Shenzhen, China. [3] Institute of Functional Nano and Soft Materials (FUNSOM), Soochow University, 215123 Suzhou, China. [4] Institute for Advanced Study, Shenzhen University, 518060 Shenzhen, China. [5] Department of Physics, The Hong Kong University of Science and Technology, Clear Water Bay, Hong Kong, Hong Kong. [6] College of New Materials and New Energies, Shenzhen Technology University, 518118 Shenzhen, China. [7] National Synchrotron Radiation Laboratory, University of Science and Technology of China, 230029 Hefei, China. [8] Department of Physics, National Laboratory of Solid State Microstructures, Nanjing University, 210093 Nanjing, China. These authors contributed equally: Shuaipeng Xing, Zhe Zhang, Xiyu Fei. Correspondence and requests for materials should be addressed to H.X. (email: xuh@sustc.edu.cn) or to J.F. (email: jianfan@suda.edu.cn) or to Y.-q.M. (email: myqiang@nju.edu.cn) or to Z.S. (email: phzshi@suda.edu.cn)

Bottom-up engineering of covalent organic nanoarchitectures via on-surface reactions has attracted intense interest in the past decade[1–8]. Both the structures and the properties of the conjugated nanostructures could be finely tuned at the single-atom level through the covalent coupling between the molecular precursors. Numerous conjugated nanostructures, including macromolecules, nanoribbons and two-dimensional (2D) polymers, have been achieved on surfaces, demonstrating great application potential in nanoelectronics, spintronics and sensing devices[9–12]. To achieve these nanostructures, a collection of covalent coupling reactions applied in traditional bulk chemistry such as dehalogenated aryl-aryl coupling and homocoupling of alkanes and alkynes have been revisited. More importantly, the insightful information of coupling reaction mechanisms has been provided at the single-atom level[13–27]. In particular, the dehalogenated aryl-aryl coupling (and cyclodehydrogenation) of predesigned precursors has succeeded in controlling the width, the structural heterogeneity and the edge topology of graphene nanoribbons (GNRs)[10,28–31]. A high on-off ratio ~$10^5$ was recorded for the field-effect transistors based on a 9-atom wide armchair GNR[10]. Recently, great advances have been made in both chemical and structural modulations of on-surface synthesized nanostructures. For instance, boron, nitrogen or sulfur-doped GNRs[32–35], porous carbon nanoribbons[36] and (doped) porous graphene[9,37] have been documented.

A control of the reaction pathway is crucial towards the precise construction of conjugated nanostructures, which still remains a great challenge, as the covalent reactions generally are of irreversibility and poor selectivity. To yield the target nanostructures rather than the side-products, the molecular precursors need to be designed delicately, and the reaction condition has to be well controlled[38–41]. In contrast, surface-confined supramolecular coordination chemistry has demonstrated the success of the highly self-recognizable, self-selective metal-organic coordination motifs (e.g., Fe/Cu–N/O bonds) in construction of desired nanostructures with the atomic scale order[42–44]. Inspired by these remarkable results, use of metal-coordination motifs as the template to control the on-surface reactions has received

attention in recent years[39,45,46]. However, a comprehensive understanding of the template effect has not been fully developed.

Herein, we demonstrate a selective on-surface covalent coupling templated by the metal-organic coordinative bonding. A nitrogen-doped porous conjugated nanoribbon structure, namely sexipyridine-phenylene (SPy-*p*-Ph)[47,48], is prepared via the debrominated aryl-aryl coupling reaction of 1,4-bis(6,6''-dibromo-[2,2':6',2''-terpyridin]-4'-yl)benzene (*p*-DBTB) on Au(111) with the Fe-terpyridine (tpy) coordination motif as the template (Fig. 1). In the absence of Fe-template the *p*-DBTB molecules with various conformations are linked together randomly to form inhomogeneous polymorphic structures on a pristine Au(111). In contrast, when the Fe atoms are introduced into the reaction system, both the molecular conformations and the C–C coupling modes are judiciously selected, leading to the formation of SPy-*p*-Ph nanoribbons upon a macrocyclization coupling. Scanning tunneling microscopy (STM), in combination with density functional theory (DFT) simulations and X-ray photoelectron spectroscopy (XPS), reveal the template effect of Fe-tpy coordination motifs on the selective on-surface covalent coupling. The generality of such a template effect is further manifested by the similar selective covalent coupling between the *m*-DBTB isomer molecules. Our approach opens an avenue for the rational design protocol towards novel conjugated nanomaterials, precisely controlled in both structure and chemistry.

## Results

**Molecular conformers and assembled structures of *p*-DBTB on Au(111).** The *p*-DBTB molecule shows excellent conformational flexibility, because its bipyridine (bpy) fragment can adopt *trans* or *cis* conformation (Fig. 1) and thus the dibromo-tpy (Br-tpy) terminal can exhibit *trans,trans*-, *trans,cis*-, and *cis,cis*- conformation. Figure 2a displays a self-assembled structure of the intact molecules adsorbed on a Au(111) surface, which was monitored with the sample held at $T = 113$ K. The molecules were evaporated onto the substrate held at room temperature (293 K). More than 90% of the molecules appear in the *trans-$D_{2h}$*

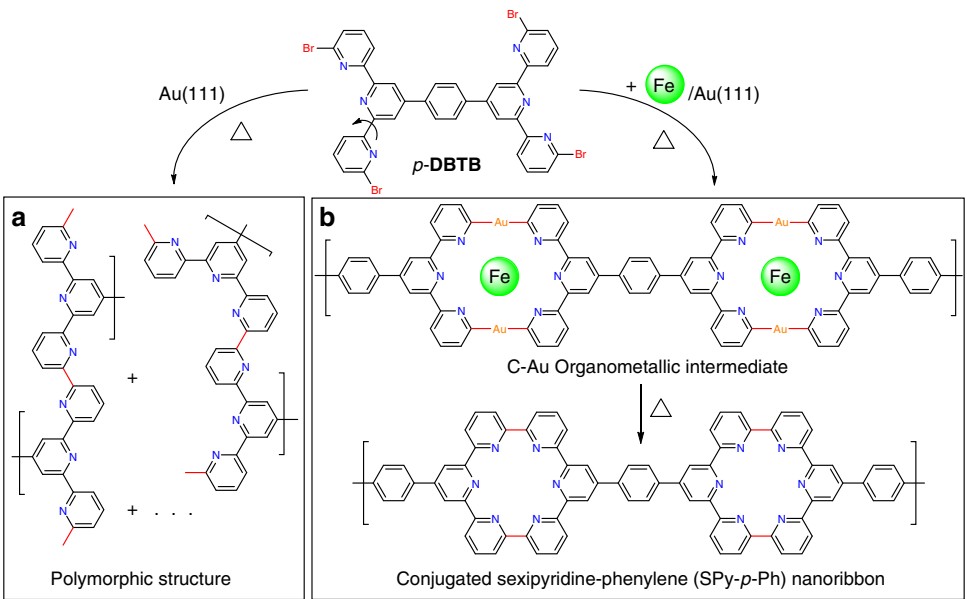

**Fig. 1** Reaction pathways of *p*-DBTB on Au(111). The precursor 1,4-bis(6,6''-dibromo-[2,2':6',2''-terpyridin]-4'-yl)benzene (*p*-DBTB) is flexible as the bipyridine (bpy) fragments can adopt *cis*- or *trans*-conformation. **a** Uncontrolled C–C coupling. The *p*-DBTB forms inhomogeneous polymorphic structures after annealing at 430–661 K, where diverse bonding modes coexist. The as-formed C–C bonds are highlighted in red. **b** Fe-tpy coordination templated macrocyclization. The template effect of the Fe-tpy coordination leads to a distinct reaction pathway, generating C-Au-C organometallic intermediates after annealing at 397–566 K and conjugated SPy-*p*-Ph nanoribbons at 578–705 K

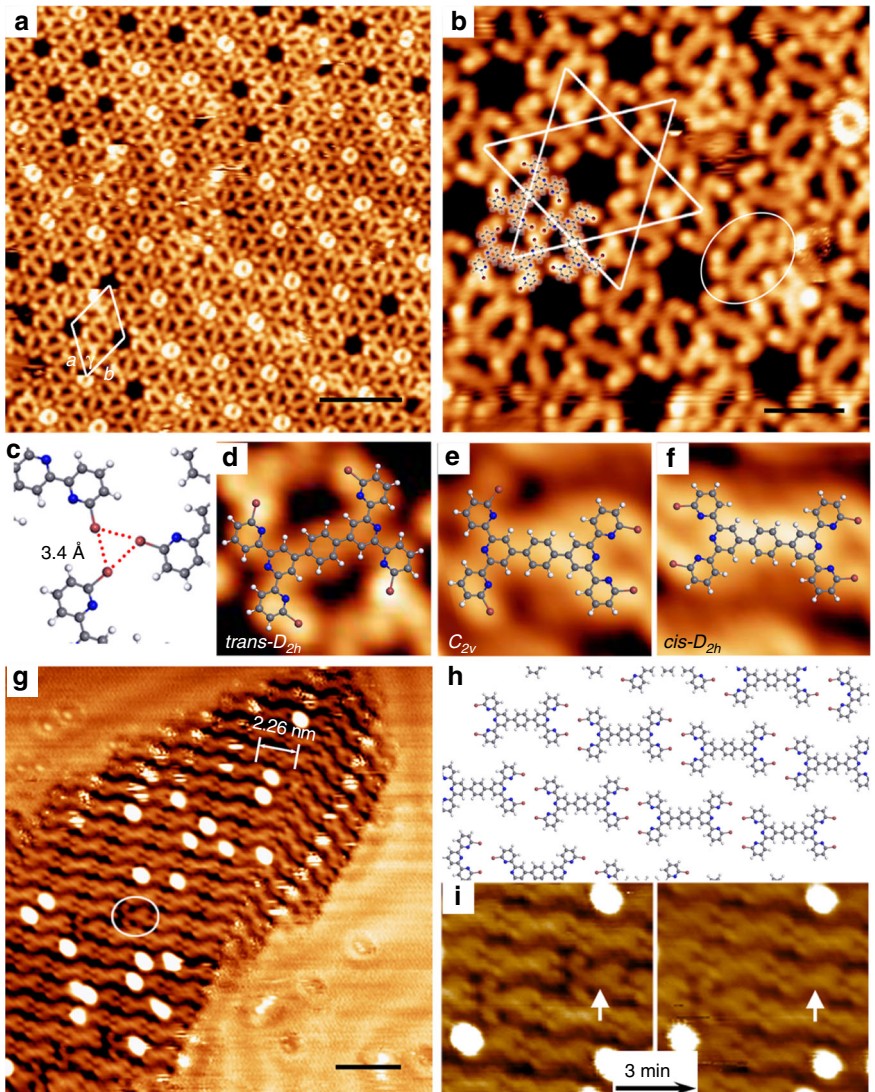

**Fig. 2** Molecular conformers and assembled structures of intact *p*-**DBTB**. **a** STM overview of the Kagome networks monitored at *T* = 113 K. A unit cell is highlighted by the rhombus (*a* = *b* = 3.35 nm, γ = 60°). **b** Most of the molecules appear in the *trans-D₂ₕ* shape. A "defect", *C₂ᵥ*-conformer, is indicated by an ellipse. Tentative structural model of a trimeric cluster is overlaid. Carbon, gray; nitrogen, blue; bromine, red; hydrogen, white. **c** Br⋯Br halogen bonds of a trimeric cluster. **d–f** Magnified topographs of the *trans-D₂ₕ*, *C₂ᵥ* and *cis-D₂ₕ* conformers, overlaid by the corresponding molecular models. **g** A close-packed island composed of *cis-D₂ₕ* conformers observed at 293 K. A "defect", *C₂ᵥ* conformer, is marked by an ellipse. **h** Tentative structural model of the close-packed structure. **i** Spontaneous conformational transition of the molecules monitored at 293 K. One *C₂ᵥ* conformers (marked by an arrow) was transformed into a *cis-D₂ₕ* conformer during a continuous scanning in 3 min. Scale bars: (**a**) 5 nm; (**b**) 2 nm; (**g**) 3 nm. Data acquisition conditions (*U, I, T*): (**a**, **b**, **d** and **e**) −1.5 V, 0.2 nA, 113 K; (**f**) −1.2 V, 0.1 nA, 293 K; (**g**, **i**) −1.2 V, 0.1 nA, 293 K

conformation (Fig. 2b, d), with both of the Br-tpy terminals adopting *trans,trans*-conformation. These molecules are assembled into a Kagome network structure, with the lattice parameters *a* = *b* = 3.35 nm, γ = 60° (Fig. 2a). The high-resolution STM image (Fig. 2b) displays that each three molecular monomers are directed 120° apart and linked together via Br⋯Br interaction to form a trimeric cluster (highlighted by molecular models). The clusters are further arranged with a head-to-head manner to form the Kagome lattice. The structural model based on the measured lattice parameters yields a Br-Br distance of 3.4 Å (Fig. 2c), falling in a typical length range of the halogen bond[2]. As shown in Fig. 2b, a small portion of molecules adopt the *C₂ᵥ* conformation (Fig. 2e), where the two Br-tpy ends of one *p*-**DBTB** molecule exhibit *trans,trans*-conformation and *cis,cis*-conformation, respectively. The donut-shaped species observed in the lattice voids could be assigned to the non-bonding *p*-**DBTB** monomers, which presented blur appearance due to the thermal rotation and

vibration[49]. The predominance of *trans-D₂ₕ* conformers at 113 K can be rationalized by the energetic preference of the molecular bpy fragments. As suggested by the theoretic calculations[50], the energy of a *trans*-shaped 2,2'-bpy molecule is 34 kJ per mol lower than its *cis*-conformer, due to the repulsion of the pyridylic N atoms and the steric hindrance from the peripheral H atoms. Thus, the *trans-D₂ₕ* conformer represents the most stable state among all the conformers.

The energy barrier of the *trans/cis* transition of the bpy fragments, however, is not sufficiently high. The *p*-**DBTB** molecules could have enough kinetic energy to overcome this rotational barrier at the elevated temperature. Figure 2g depicts a typical close-packed domain composed of chain-like structures when the sample was held at room temperature (*T* = 293 K). Interestingly, most molecules (>77%) in the domains appear as *cis-D₂ₕ* conformers (*i.e.*, both of the Br-tpy terminals of a molecule adopt *cis,cis*- conformation; see Fig. 2f). These molecules

are interlinked in a head-to-head manner along the main axis of the molecule. The measured center-to-center distance between two adjacent molecules is 2.26 nm. The tentative structural model (Fig. 2h) reveals a distance of 4.2 Å between two closest Br atoms (belonging to the adjacent molecules, respectively). There also are some "defects", $C_{2v}$ conformers, in this domain (marked by an ellipse in Fig. 2g). Figure 2i depicts a spontaneous transition from a $C_{2v}$ conformer to a $cis$-$D_{2h}$ conformer during a consecutive scanning in 3 min. This transition observed at 293 K manifests unambiguously the conformational flexibility of the $p$-**DBTB**. Note that the chain-like structures were loosely packed and subject to dissolve during the scanning. Thus, there existed a large amount of gas-phase molecules, whose conformations were unknown.

However, such excellent conformational flexibility of $p$-**DBTB** results in a poor controllability in the reaction pathway towards forming the desired conjugated nanostructures; particularly the reaction is to take place at even elevated temperatures via the debrominated aryl-aryl coupling.

### Debrominated aryl-aryl coupling of $p$-DBTB in the absence of Fe

The thermal annealing at 430 K initiated the debrominated aryl-aryl coupling of the molecules, yielding an inhomogeneous polymorphic structure. The polymorphic structures remained intact when the annealing temperature was increased to 661 K (Fig. 3a), indicating that $p$-**DBTB** molecules (after debromination) were covalently linked together. Close-up inspections reveal that the structure is composed mainly by rectangular- and triangular-enveloped sub-structures. Small dots are discernible inside these sub-structures, which can be assigned to the cleaved Br atoms. Figure 3b–e provide the typical topographs of both sub-structures with the structural models. The length and width of the rectangular sub-structure are 2.27 nm and 1.17 nm, respectively. The four sides appear in the similar contrast, reflecting a flat molecular adsorption. This rectangular sub-structure is constructed by $trans$-$D_{2h}$ $p$-**DBTB** molecules, which are jointed together with a $trans$-$trans$ bpy-bpy connection (Fig. 3c). The length (2.25 nm) and width (1.15 nm) of the structural model fit well with the STM results. Similarly, the structural model of the triangular sub-structure involves $trans$-$cis$ bpy-bpy connections (Fig. 3e), and the length of the lateral side is 1.87 nm. The value agrees well with the STM result (1.89 nm, Fig. 3d). Notably, a $cis$-$cis$ bpy-bpy connection (Fig. 3f) produces a small hexagonal macrocycle, where two $cis$,$cis$-tpy terminals are linked in a head-

to-head manner. The center-to-center distance between two molecules measures 1.75 nm, which is very close to the phenyl-to-phenyl distance based on the simulation model (1.73 nm, Fig. 3g). Thus the hexagonal unit can be identified as a covalently coupled structure, namely sexipyridine (SPy)[47,48]. As expected, this SPy unit was rarely detected, because neither the $cis$,$cis$-tpy conformation nor the macrocyclization reaction was thermodynamically favored. Note the C-Au-C intermediates were not observed under this reaction condition, in line with the previous reports[3,13].

The structural inhomogeneity of the polymers in this reaction could result from the existence of manifold molecular conformers and various C–C coupling modes between the molecules. Figure 3h provides the statistics (by counting ~290 molecules) on the conformations of the tpy moieties. Considering the energetic preference for $trans$-bpy fragment[50], it is not surprising that the $trans$,$trans$-tpy terminal dominates within the polymorphic structure (64%). Figure 3i shows the analysis of the as-formed C-C coupling modes (~327 modes) of the bpy-bpy connections. Three different coupling modes coexist, because the covalent coupling is non-selective. The $trans$-$trans$ C–C coupling mode is the major reaction pathway (78%), for the $trans$-bpy being energetically favored. Nevertheless, the thermal annealing reaction condition gives poor reaction selectivity and leads to multiple coupling modes (assisted with various molecular conformers), due to the presence of various reaction pathways as illustrated in Fig. 1.

### Fe-tpy templated C–C coupling–organometallic intermediates

To control the reaction pathway of this on-surface reaction, the Fe atoms were introduced into the reaction system to regulate both the molecular conformations and the C–C coupling modes (see Fig. 1). It has been well documented that the Fe atom/ion shows strong coordination interaction with tpy ligand both in solution and on surfaces[49,51–54]. The theoretically estimated binding energy of Fe-tpy up to ~287 kJ per mol[55] is expected to afford a good template effect, with which the conformation and the coordinative/covalent coupling mode of the (Br-)tpy entities can be selected.

Following the codeposition of the $p$-**DBTB** molecule with Fe atoms on a Au(111) substrate held at 293 K, the sample was cooled down to 100 K for STM measurements. A typical STM overview (Fig. 4a) manifests an assembled structure dramatically distinct from what was observed in the absence of Fe. Most

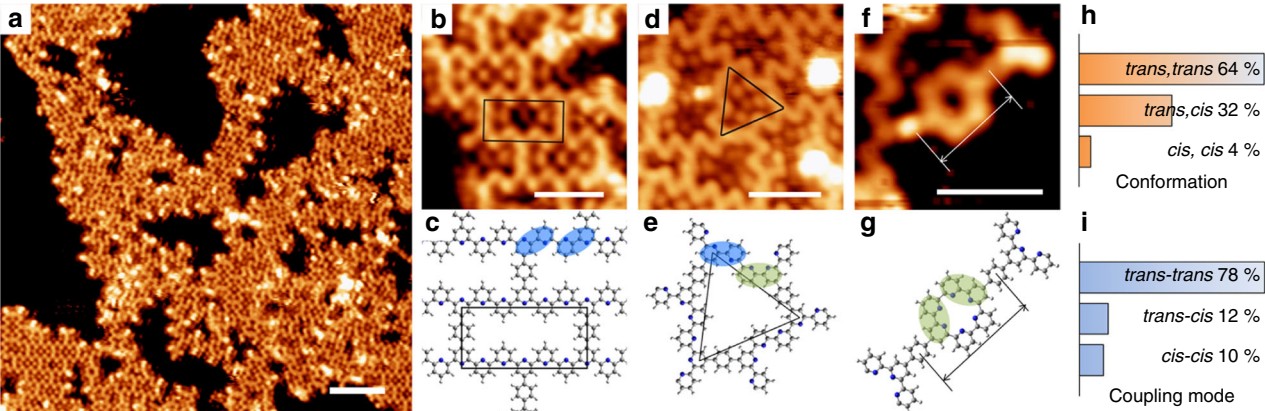

**Fig. 3** Uncontrolled debrominated aryl-aryl coupling reaction. **a** STM overview of the sample after annealing at 661 K. **b–g** Three typical conjugated sub-structures and the corresponding structural models. The conformations of bpy fragments are highlighted by ovals in different colors ($trans$, blue; $cis$, green). Different as-formed bpy-bpy connections are observed, namely $trans$-$trans$, $trans$-$cis$ and $cis$-$cis$. **h** Statistic analysis of the molecular conformers (based on the conformations of tpy terminals). **i** Statistic analysis of the three typical C-C coupling modes of the bpy-bpy connections. Scale bars: (**a**) 5 nm; (**b**, **d**, and **f**) 2 nm. Data acquisition conditions ($T$ = 293 K): (**a**) −1.0 V, 0.4 nA; (**b**) −0.5 V, 0.2 nA; (**d**) −0.6 V, 0.15 nA; (**f**) −0.5 V, 0.08 nA

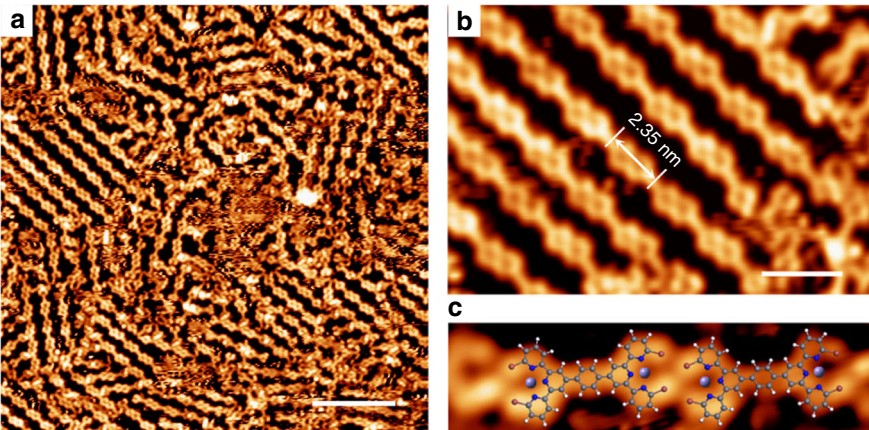

**Fig. 4** Molecular conformers and assembled structures in the presence of Fe. **a** STM overview of the chain structures observed when the sample held at 100 K. **b** Close-up inspection of the chains. Most molecules are *cis-$D_{2h}$* conformers. **c** Tentative structural model. Fe coordination centers are represented by purple spheres. Scale bars: (**a**) 10 nm; (**b**) 3 nm. Data acquisition conditions ($T = 100$ K): (**a**, **b** and **c**) −1.8 V, 0.2 nA

molecules (>90%) appear in *cis-$D_{2h}$* conformation, link to each other with their Br-tpy terminals, and form a chain structure. The close-up inspection (Fig. 4b) reveals a center-to-center distance of 2.35 nm between neighboring molecules within a chain. Although the Fe-coordination centers were not resolved in the STM topographs[49,56], the dominant observation of the *cis-$D_{2h}$* conformers could be attributed to the coordination interaction between Fe and the *cis,cis*-Br-tpy moieties, because the binding energy (287 kJ per mol) is much higher than the *trans/cis* transition energy barrier of bpy fragments (34 kJ per mol). The tentative model (Fig. 4c) deduces a distance of ~3.8 Å between the nearest Br atoms (Supplementary Fig. 1), suggesting that a typical halogen bonding[2] is responsible for the formation of the chains (Supplementary Note 1). Note that these assembled chains became unstable and were partially dissociated at room temperature, and only short chains or small molecular clusters were resolved occasionally (see Supplementary Note 2 and Supplementary Fig. 2). So the preorganization of the molecules (at $T = 100$ K) is unlikely to guide the next coupling reaction that takes place at higher temperature[57].

To invoke the debrominated C-C coupling reaction, the sample was annealed to 397-566 K. STM reveals the arrayed domains composed of linear chains that are distributed over the entire surface (Fig. 5a). The chains extend along the crystallographic directions < 1–10 > of the substrate, and most are further packed into domains. A close-up inspection (Fig. 5b) reveals that *p*-**DBTB** molecules are interlinked in a head-to-head manner within a chain. The center-to-center distance between neighboring molecules measures $d = 2.04 \pm 0.02$ nm, displaying a perfect commensuration with the substrate < 1–10 > vectors ($a = 2.88$ Å); $d$ is about $7a$ (2.016 nm). Three linearly aligned "spots" (marked by symbols + or ×, Fig. 5b) are discernible in the links (labeled L-mode) between neighboring molecules with the alignment perpendicular to the chain. Some outmost spots (+) are missing as indicated by the black arrows in Fig. 5b, while the white arrow points out a tpy terminal without Br atoms. The small dots (○) discernible in the inter-chain spaces (Fig. 5b) can be assigned to Br atoms, which are cleaved from molecules. The ratio of Br/molecule within the domain is 4, suggesting all Br atoms being cleaved. This assignment is also supported by our complementary XPS measurements (Supplementary Note 3), where only the chemisorbed Br atoms were identified in the similar reaction condition (Supplementary Fig. 3). Thus, we can conclude that the debromination reaction has taken place under this annealing condition. Accordingly, the "+" spots in Fig.5b could be

assigned to Au atoms that linked *p*-**DBTB** molecules together via C-Au-C organometallic bonds. Based on the dimensions of the chains ($d$) and of the molecules, the calculated C-Au organometallic bond length is 2.2 Å, fitting well with the previous reports[13,58].

The high-resolution STM image acquired with a very small bias voltage ($U = -0.05$ V, Fig. 5c) revealed more details for the C-Au-C organometallic chains. The topograph indicates that there is one additional spot (white "×") located beside the central spot (black "×") and in the bay of each tpy terminal. The recent reports have presented similar Fe-tpy coordination chains on Ag (111)[53,54], where the Fe trinuclear cluster was unambiguously demonstrated. Thus, we tentatively assign the three "×" spots within an L-mode link to three linearly distributed Fe atoms. The central Fe atom exhibits a bright feature relative to other two Fe atoms, distinct from the scenario on Ag(111)[53,54], where the central Fe atom displayed a relatively dim contrast. To interpret this feature, we have carried out DFT calculations (see Supplementary Note 4 and Supplementary Fig. 4). The DFT-optimization results suggest that there is one Au atom under the central Fe atom, which is lifted upwards above the molecular plane (Fig. 5d, e). The simulated STM topograph ($U = -0.05$ V, Fig. 5f) displays a bright protrusion on the central Fe position, fitting well with the major feature revealed in STM observations.

In this reaction pathway, the organogold chains emerged as intermediates, which were not produced with the same reaction precursors in the absence of Fe (that otherwise generated the C–C coupled polymorphic structures). In previous reports[30,58], the stable organogold intermediates were observed, generally because the next C–C coupling step was inhibited by the great steric hindrance between the adjacent reactive organic units. In our system, the formation of Fe-coordination cluster leads to a relatively large separation between two *cis,cis* tpy terminals. Thus, the C–Au–C intermediate cyclic structure can be exclusively generated, because it leaves a larger void than the cyclic C–C coupled product does. Notably, the acyclic organometallic connection[38,59] of the *cis,cis* tpy terminals was not observed in this phase. As illustrated in Fig. 1, these results manifest that the Fe-tpy coordination motifs not only control the molecular conformation but also guide chemical reaction pathway towards the cyclic organometallic products.

**Final porous N-doped SPy-*p*-Ph nanoribbons.** Thermal annealing at 578–705 K invoked the last step of the debrominated

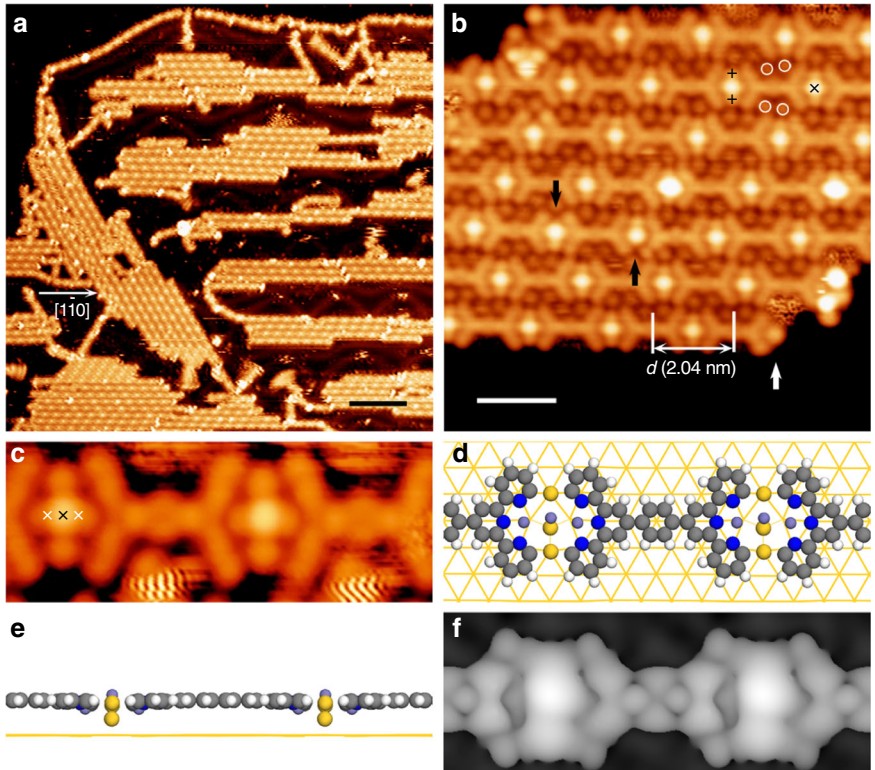

**Fig. 5** C-Au organometallic intermediates after annealing at 397–566 K. **a** Overview of the organometallic chains after annealing at 487 K. **b** Close inspection of the chain structure. Black arrows denote the missing of Au atoms. The white arrow marks a debrominated tpy end. Three types of spots are marked by different symbols (plus sign +, cross sign × and circle ○). **c** High-resolution STM image ($U = -0.05$ V) of the chains ($4.1 \times 1.6$ nm$^2$). Additional Fe atoms, as marked by white "×", are resolved. **d–f** DFT optimized structure of the organometallic chain: top view (**d**), side view (**e**) and simulated STM topograph ($U = -0.05$ V) (**f**). Fe, purple; Au adatom, yellow; C, gray; H, white. Scale bars: (**a**) 10 nm; (**b**) 2 nm. Data acquisition conditions ($T = 293$ K): (**a**) −1.2 V, 0.14 nA; (**b**) 0.3 V, 0.05 nA; (**c**) −0.05 V, 0.1 nA

aryl-aryl coupling reaction, leading to the formation of a pure organic C-C coupled ribbon structure. Figure 6a depicts an STM overview of the sample with the annealing treatment at 705 K. The discrete ribbons as long as 50 nm are formed with a preferred orientation along the substrate vectors < 1−10 >. These ribbons show excellent structural strength, as suggested by the fact that some chains can freely vibrate on the surface while maintaining their structures. The measured center-to-center distance between two neighboring molecules within the ribbons is 1.75 ± 0.05 nm (Fig. 6b). This value fits well with the DFT-optimized sexipyridine-p-phenylene (SPy-p-Ph) molecular model (1.73 nm, Fig. 6c, middle). The simulated STM topograph ($U = -0.5$ V, Fig. 6c, right) also agrees well with the experimental observation, showing the similar porous hexagonal knots. As shown in Fig. 6c–e, the high-resolution topographs reveal three different knots based on the contrast of their central parts (I, dark; II, dim; III, bright; also see Fig. 6b). Despite the different contrasts at the knots centers, all of the knots are hexagon-shaped, demonstrating that all of them are the SPy unit built through the macrocyclization of two cis,cis tpy terminals. Therefore, we conclude that the C-Au-C organometallic chains have converted into a pure organic SPy-p-Ph nanoribbon structure, as illustrated in Fig. 1.

To our best knowledge, this SPy-p-Ph structure is the first example of the porous nitrogen-doped carbon nanoribbons fabricated on surfaces. Our DFT calculations (Supplementary Note 5) on the band structure of the ribbon predict a direct bandgap of 2.3 eV. The charge density of valence band maximum (VBM) and conduction band minimum (CBM) are spatially separated (Supplementary Fig. 5). The VBM is mainly localized at

N atoms of the SPy moieties, while the CBM is primarily distributed at C atoms of Ph moieties. Such band structure could allow the SPy-p-Ph nanoribbons to be used in organic optoelectronic nanodevices[60].

We note that SPy is one type of torand molecules, which can host metallic ion/atoms through non-covalent interactions[61]. Thus, the SPy units (I-mode) could accommodate Fe atoms (II-mode), or Fe-related clusters (III-mode), and thus exhibit different contrasts on the central parts; see Supplementary Note 6 and Supplementary Fig. 6 for the theoretical simulations. Stepwise postannealing treatments to a sample at 650, 673, and 705 K (Fig. 6f) revealed an increment in the abundance of I-mode with the decrement of II/III-modes as the annealing temperature increased. This tendency can be interpreted as a result of thermal annealing process where the non-covalent bond between the guest and SPy macrocycle is broken at high temperature.

**Generality of Fe-tpy coordination templated C–C coupling.** To demonstrate the generality of this Fe-tpy coordination templated covalent coupling, we have conducted experiments using the molecule m-**DBTB** (see the inset of Supplementary Fig. 7a for its chemical structure). In contrast with the inhomogeneous polymorphic structures in the absence of Fe (see Supplementary Note 7 and Supplementary Fig. 7), the thermal annealing treatment (at 457 K) to a Au(111) sample codeposited with m-**DBTB** and Fe led to the formation of zigzag-chain arrays (Fig. 7a). Our structural analysis demonstrates that the zigzag-chains contain both C–Au–C intermediates (Fig. 7b) and C–C coupled SPy units (Fig. 7c). Notably, a large number of the III-mode SPys have been

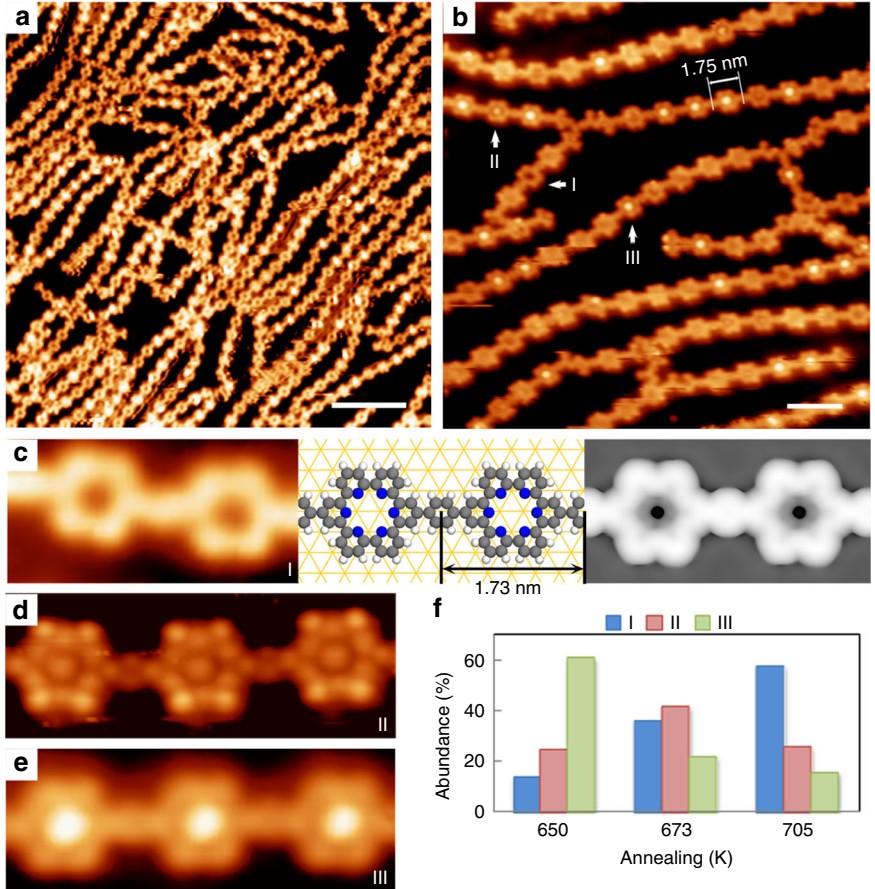

**Fig. 6** SPy-*p*-Ph nanoribbons formed after annealing at 578-705 K. **a** STM overview of a sample annealed at 705 K. **b** Close-up inspection of the nanoribbons. The measured periodicity of the chain is 1.75 ± 0.05 nm. Labels I, II and III denote the three modes for the SPy macrocycles. **c** High-resolution STM image (left), DFT-optimized structure (middle) and the simulated STM topograph (right, $U = -0.5$ V) of the I-mode SPy. **d, e** High-resolution STM images of the II and III-mode. **f** The dependence of the abundance of I, II and III-mode SPy units on the annealing temperatures (650 K, 673 K and 705 K). Scale bars: (**a**) 10 nm; (**b**) 3 nm. Data acquisition conditions: (**a**) −0.8 V, 0.2 nA, 105 K; (**b**, **c** and **e**) −0.5 V, 0.2 nA, 133 K; (**d**) −1.5 V, 0.3 nA, 293 K

observed at this annealing temperature relative to the other two modes of SPys, which is in agreement with the tendency revealed in Fig. 6f. As expected, all molecules became covalently coupled via the macrocyclization between their tpy terminals with the annealing temperature at 547 K (Fig. 7d–f). The resultant SPy-*m*-Ph nanoribbons extend on the surface in a meandering manner, due to the chevron-shape of the molecular precursors. Interestingly, the C–Au–C organometallic and C–C coupled hyper-ring units, each was composed of six *m*-**DBTB** molecules, were observed with annealing temperature at 457 K and 547 K, respectively (Fig. 7g–j). The C–C coupled organic hyper-ring has a diameter as large as 3.0 nm, representing a macromolecular compound. We propose that the pure phase of this hyper-ring structure can be obtained with the pseudo-high-dilution approach[38].

## Discussion

To sum up, we propose a tentative scenario to describe the template effect of the Fe-tpy coordination as the following: (i) The Fe-tpy coordination converts most of molecular precursors into *cis-D*$_{2h}$ conformers, due to the large binding energy between Fe and tpy unit and the excellent conformational flexibility of the molecules. (ii) The linear geometry of tpy-Fe-tpy coordination motifs guides the reaction pathway exclusively towards the organometallic L-mode chain structures. (iii) The gradual decomposing of Fe clusters within the macrocycle structure at

high temperatures allows the "contraction" of the C–Au–C intermediates to finalize the C–C coupling reaction.

In conclusion, we have demonstrated an excellent template effect of Fe-tpy coordination on guiding the reaction pathway of a C–C coupling between the tpy derivatives. This Fe-tpy coordination templated reaction opens up prospects for synthesizing a series of 1D porous polymers, or even 2D networks facilely with tpy derivatives as precursors. Moreover, based on the general principles delineated here, other molecular precursors with specific functional ligands (like bpy) are expected to react in a controlled manner via an appropriate metal-organic coordination motif as the template. Our approach opens an avenue for the rational design and synthesis of 2D functional nanomaterials at the single-atom level.

## Methods

**Synthesis**. The molecular precursors *p*-**DBTB** and *m*-**DBTB** were synthesized according to published procedure[62] by the condensation reaction between corresponding enones and pyridinium salt in the presence of ammonium acetate.

**STM**. Sample preparation was performed in an ultrahigh vacuum (UHV) system (SPECS GmbH) at a base pressure < $3.0 \times 10^{-10}$ mbar. The single-crystal Au(111) substrates (MaTeck, 99.999%) were cleaned by cycles of Ar ion sputtering at an energy of 900 eV and annealing at 800 K. The cleanness of the pristine Au(111) samples was determined by STM scanning at room temperature that produced atomic-resolution topographs. Fe atoms were evaporated from an iron rod (Puratronic, 99.995%) by using an electron-beam evaporator. The molecular precursors were evaporated by organic molecular beam epitaxy (DODECON Nanotechnology

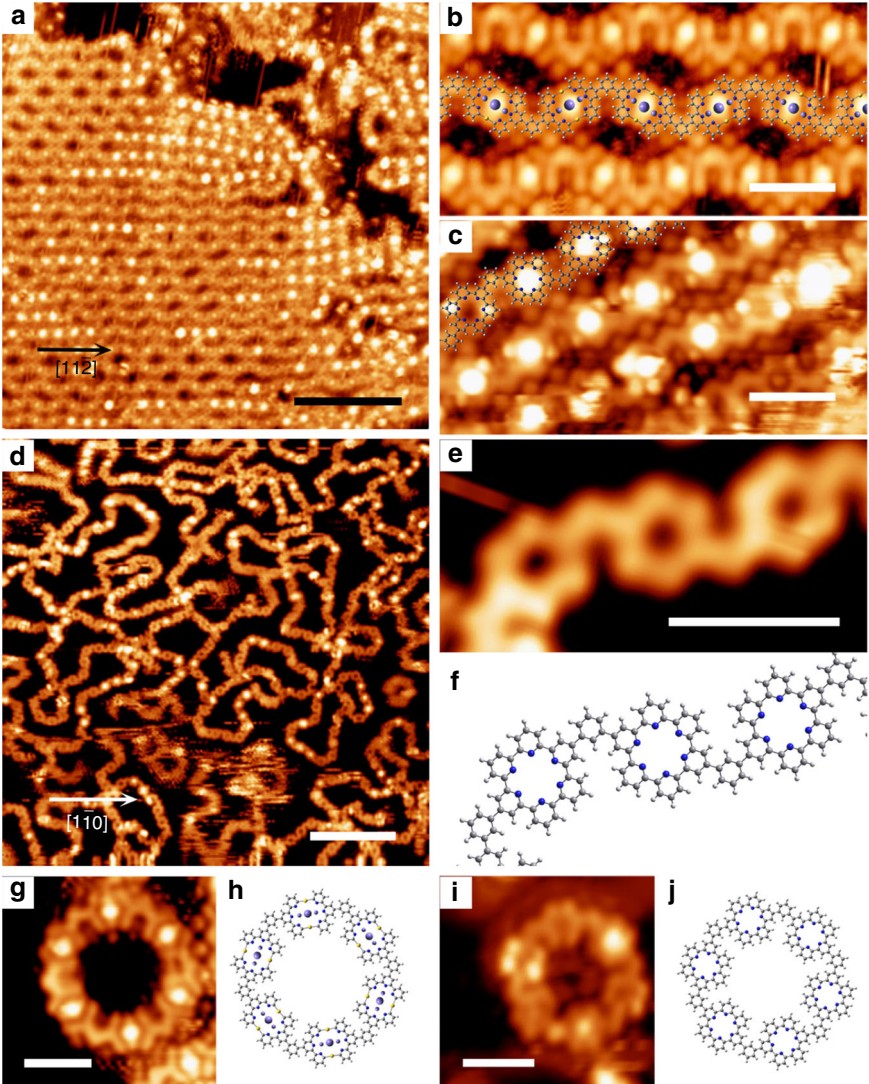

**Fig. 7** Fe-tpy templated debrominated C–C coupling of *m*-**DBTB**. **a** An STM overview with the sample annealed at 457 K. **b** High-resolution STM of the organometallic chains with structural models overlaid. The Fe-coordination is represented by purple spheres, while the central sphere is bold to highlight the specific state of the central Fe atom. **c** High-resolution image of SPy units (III-mode). **d** STM overview of the SPy-*m*-Ph ribbons after annealing at 547 K. **e**, **f** Segments of the ribbon and the structural model. **g–j** The C-Au-C organometallic and C-C coupled hyper-ring units with their corresponding structural models. Scale bars: (**a**, **d**) 10 nm; (**b**, **c**, **e**, **g** and **i**) 2 nm. Data acquisition conditions ($T = 293$ K): (**a**) −1.0 V, 0.5 nA; (**b**) −0.5 V, 2.0 nA; (**c**) −1.2 V, 0.5 nA; (**d**) −1.0 V, 0.2 nA. (**e**) −1.0 V, 1.2 nA; (**g**) −1.0 V, 1.0 nA; (**i**) −1.0 V, 0.2 nA

GmbH), and the sublimation temperature was 405 °C (*p*-**DBTB**) and 400 °C (*m*-**DBTB**). All STM experiments were performed using an Aarhus SPM apparatus controlled by the Nanonis electronics. Topographic data were acquired in constant current mode, with the bias voltages applied to the sample, and with the sample temperature at 100–293 K (room temperature). The STM images were processed by using the software WSxM[63].

**DFT calculations**. The spin-polarized calculations were performed using the Vienna Ab initio Simulation Package (VASP)[64] based on density functional theory (DFT)[65] within the Perdew-Burke-Ernzerhof (PBE) exchange-correlation functional[66]. The projector augmented wave (PAW) method[67] was used to describe the interactions between ions and electrons. A plane-wave-basis with kinetic-energy cutoff of 500 eV was used. The geometry of molecular monomers were obtained by fully relaxing the atomic positions of a single molecule in a $32 \times 32 \times 21$ Å vacuum region. In periodic conditions, the vacuum between two molecules is at least 15 Å. To simulate the molecules on the surface, a three layer of Au(111) slab with the bottom layer fixed was built. The vacuum region was larger than 12 Å to avoid interactions between neighboring images. A $p(10 \times 7)$ supercell consisting of 210 Au atoms was employed to simulate the organometallic chain configuration, while another $p(9 \times 6)$ supercell consisting of 162 Au atoms for SPy and its II/III modes was employed. All atoms were fully relaxed until the forces acting on each atom

were less than 0.02 eV per Å. The DFT-D3[68] method of Grimme was used to evaluate the van der Waals (vdW) effect. To investigate the correlation effect of Fe $3d$ electrons, the GGA + U method with the effective Coulomb energy $U = 3$ eV was used.

**XPS measurements**. Following the sample preparation performed in our UHV system in Suzhou, the sample was transferred, by using a sample transfer chamber (< $5.0 \times 10^{-8}$ mbar), to the Catalysis and Surface Science Endstation at the BL11U beamline in the National Synchrotron Radiation Laboratory (NSRL) (Hefei, China)[69] for the XPS experiments. The C 1s and Br 3d XPS spectra were measured with a photon energy of 380 eV. The Br 3d spectra deconvolution was performed using the XPS Peak 41 program with Gaussian functions after subtraction of a Shirley background.

### Data availability
The data that support the findings of this study are available from the article and Supplementary Information files, or from the corresponding authors upon request.

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

## Acknowledgements

This work was financially supported by the NSFC (21303113, 21472135, 11474155, 11774147, 91427302), the Natural Science Foundation of Jiangsu Province (BK20130285, BK20151216) and the China Postdoctoral Science Foundation (2013M540460). The work was also funded by Collaborative Innovation Center of Suzhou Nano Science and Technology (CIC-NANO), Soochow University and by the Priority Academic Program Development of Jiangsu Higher Education Institutions (PAPD), and the 111 Project. Z.L.S. thanks Dr. Shiyong Wang for fruitful discussion.

## Author contributions

Z.L.S., J.F. and Y.Q.M. conceived and designed the project. S.P.X., R.Z., T.L. and Z.L.S. performed STM measurements, analyzed data and prepared the manuscript. Z.Z. conducted theoretical calculations under the supervision of H.X., X.Y.F. and D.L.Z. carried out the synthesis of the precursor molecules under the supervision of J.F., W.Z., H.X.J. and J.F.Z. performed XPS measurements and analyzed the data. All authors discussed the results and contributed in the manuscript preparation.

## Additional information

**Competing interests:** The authors declare no competing interests.

