## [Peer Review File · Nature Communications]

Reviewers' comments:

Reviewer #1 (Remarks to the Author):

This manuscript describes the fabrication of N-doped porous carbon nanoribbons. The authors have designed control experiments to highlight the template effect of Fe-tpy coordination, which can control the on-surface polymerization to form linear nanoribbons. Each section is discussed in detail and shows a complete process. However, analogical on-surface polymerization with metal-coordinated template has been intensively reported by Lin's group (*J. Am. Chem. Soc.* 2013, 135, 3576, etc.). Not long ago, Aitor Mugarza et al. also reported bottom-up synthesis of nanoporous graphene by on-surface aryl-aryl coupling reaction and its electronic properties (*Science*, 2018, 360, 199). In terms of the criteria of publication with Nature communications, this manuscript lacks of deep insight and novelty, hence I could not recommend its publication. Detailed comments are shown below.

- 1) In Figs.1a and b, the STM image of the Kagome lattice show that the "hexagonal pores" depict either dark or bright central protrusions. What is the difference between them?
- 2) Line 18, page 1, the expression "debromination aryl-aryl coupling reaction" seems weird and should be replaced by "debrominated" or "aryl-aryl coupling reaction after debromination". Such an expression is used frequently in the text.
- 3) Line 21, page 1, "yielding the pure phase of SPy-p-Ph porous nanoribbons" is skeptical because according to the text, the ultimate products based on SPy-p-Ph have three modes.
- 4) Line 25, page 4, "The high-resolution STM image provided in Fig. 1b displays that three molecular monomers tilted by 60° are linked together via Br...Br interaction to form a trimeric cluster." The Br...Br interaction as well as the formed trimers should be marked in Fig. 1b.
- 5) Line 10, page 5, "The theoretically estimated torsional barrier for the trans/cis transition of a 2,2'-bpy molecule in free space is about 34 kJ/mol, much higher than that for the cis/trans transition ~6 kJ/mol⁴⁹." This sentence is in contradiction to the reference whose Fig. 2 actually includes the thermodynamic information which supports that the trans configuration is more stable.
- 6) Line 12, Page 6, "The thermal annealing at 157 °C initiated...an inhomogeneous polymorphic structure (Fig. 2a). " Meanwhile, the caption of Fig. 2a is "STM overview of the sample following an annealing treatment at 388 °C. " What is the real temperature then?
- 7) Line 5, page 8, "The Fe was used for its capability to coordinate with the tpy endgroups, which has been well documented both in solution and on surfaces⁵⁰⁻⁵⁴." But Ref. 52 only reports the assembly of the cis-BTP-TPE molecules with the tpy group on Cu(111), and has nothing to do with Fe. Such a citation is redundant.
- 8) Line 8, page 9, the digital "3" in "Fe₃ clusters" should a subscript.
- 9) Line 3, page 10, "The image indicates two extra 'spots', each in the 'bay' of the tpy terminals." This sentence is unclear because there are too many bright spots in Fig. 4c. The authors should clarify the two specific spots and mark them in the figure.
- 10) Line 2, page 15, the authors mention in the experimental section "with the sample temperature at 100 K–293 K". The specific temperature should be indicated in each figure to judge the imaging quality and the surface states. In Line 3, "The STM images were processed by using the software

WSxM" should provide the reference: Horcas, I. et al, WSXM: A software for scanning probe microscopy and a tool for nanotechnology. Rev. Sci. Instr. 78, 013705 (2007).

- 11) There are impurity molecules in the sample, making the image quality in Figs. 1 d and f poor.
- 12) In Fig. 1f, the C2v molecule marked by the white arrow in the left panel does not transform into the cis-D2h isomer after image scanning, which should be pointed out in the text.
- 13) The caption for Figs. 2e-f is ambiguous for the statistical analyses of the molecular conformations and coupling modes.
- 14) The imaging temperature should be lowered to increase the STM image resolution. For instance, the coordinating Fe atoms could not be discerned in Fig. 3.
- 15) More data should be provided to evidence the formation of the Fe₃ clusters and the enantiomeric F atom positions in II/III mode enantiomers in Fig. 4.
- 16) In Fig 4a, the marked surface lattice orientation is wrong. It should be [1-10] rather than [110] because the latter does not exist on Au(111).
- 17) Some molecules in Fig.S1 seem to have already formed the structures similar to those Fig. 4, which is different from description in the text.
- 18) In Fig. 5d, the bias of high-resolution STM images is -1.5 V which is different from that in Fig. 5b (STM overview, -0.5V). Please explain why not choose the same bias?
- 19) In Figs. 4 and 5, the structural analyses involving the attribution of the bright spots between the chains or inside the cycles are not convincing merely by the calculated simulations. Such an attribution is rather weak: i) How to attribute the central spots for the L-mode link in Fig. 4c and the bright spots inside the SPy in Figs. 5d and 5e to the Fe or Au atoms? ii) To explain the big central bright spot in the L-mode link, the authors add an Au adatom beneath the central Fe atom in their calculation (Fig. 4d) which does not agree with experimental results in Fig. 4c.
- 20) In Fig 6g, the products are not in agreement with the modelled structures in the right panel.
- 21) The measuring distance should be unified, i.e., the center-to-center distance between neighboring molecules.
- 22) The unit of the temperature should be unified by using either K or °C.
- 23) It's suggested that the co-existing products in the structures over the transition temperature ranges should be provided to support the proposed reaction pathways.
- 24) The authors should explain in detail why the introduction of the Fe atoms could change the reaction pathways to form the organometallic C-Au-C intermediates.
- 25) The band gap of GNRs makes it an attractive semiconductor material for its quantum confinement. The authors may explore in detail the electronic properties of the SPy-p-Ph, which would definitely make this work more striking.

26) This text is full of typos and grammatical flaws, which makes the text hard to read. Some examples are listed below, to just name a few.

Line 4, Page 2: "bulky chemistry" should be "bulk chemistry".

Line 17, Page 2: "coupling generally are..." should read as "coupling generally is...".

Line 17, Page 2: " has" should be " have".

Reviewer #2 (Remarks to the Author):

The manuscript under review reports on the on-surface Ullmann coupling on Au(111) assisted by a coordination template. Specifically, a precursor with two terpyridyl units and four Br is used. The presence of Fe atoms as a template-forming metal center leads to the formation of oligomer chains with regular N-heteromacrocyclic repeat units. In contrast, a disordered covalently bonded phase is obtained in the absence of the template metal. This manuscript is an excellent example for a strategy that used reversible bonds for structure formation and induces the formation of (here irreversible) covalent bonds in a successive step. By using a second precursor with a different geometry, the authors confirm that their approach is not limited to one successful example. While the formation of the final covalent products is well confirmed by the STM images, the discussion of the intermediates raised some serious questions, that can easily be addressed by additional measurements:

1. Fe-tpy templated molecular phase: Figure 3 shows STM images of the presumably intact p-DBTB precursor with Fe on Au(111). It is stated that the images were acquired with the sample held at 100 K. However, I could not find any information about the highest temperature to which the sample was exposed. This is important, because Fe is likely able to dissociate the C-Br bond already at a lower temperature than the gold surface does. This point is especially crucial to explain the bonding situation in the chains in Figure 3 (see also next comment). XPS should be used to clarify whether the C-Br bonds are still intact or not.

2. To form the proposed halogen bonds in Figure 3c, the Br atoms (or, rather, the C-Br bonds) have an unfavorable angle. Normally, angles between 120° and 90° are observed, whereas in Figure 3c the this angle is in the range of 150°-160°. At some point, the interaction should become repulsive because of the predominant interaction of the sigma-holes. Are there examples for such large angles in the literature?

3. Fe-tpy templated organometallic phase: The postulated bonding motif with 3 Au adatoms and 3 Fe atoms is very unusual, especially as the central Fe atom is very much elevated. There should be other, more convincing explanations. Could the bright features in Figure 4b represent iron bromide clusters? Considering that bulk iron bromide is much more stable than bulk gold bromide, one should expect that the Br adatoms bind preferentially to Fe. This question can be addressed by XPS investigations, which provide details about the chemical state of Fe and Br on the surface.

4. Conjugated SPy-p-Ph nanoribbons: Some of the knots (pores) in the ribbons contain Fe centers, while others are empty. The authors explain this with a tendency of Fe to diffuse to the bulk and cite literature for metal-organic networks. However, there the presence of Br could change the situation here. Perhaps the Fe centers can be more easily released because they bind strongly to Fe? Are there any possibly FeBr_x related features visible in the STM images? Again, XPS could show whether there is Br on the surface after the annealing procedure or not. With the Fe on the surface, Br may desorb at much higher temperature than from "clean" (i.e., Fe free) gold.

Reviewer #3 (Remarks to the Author):

In this manuscript, Xing et al report a study of using coordination template effects to achieve homogeneous holey nanoribbons. By comparing with the control experiments without using Fe, they demonstrated that the template play critical roles to control on-surface synthesis. Moreover, they used another precursor molecule with similar terpy functions to prove this concept is generally applicable in on-surface synthesis. The data are in good quality and very convincing. Template effects are well known in organic synthesis. To my view, this work is the first work of successfully utilizing coordination as template to control on-surface reactions. It is an exciting breakthrough in the rapidly developed field of on-surface synthesis. I this work will be highly appreciated in this community and thus strongly support its publication in Nature Communication after some minor revisions being made addressing the comments listed below.

Comments:

(1) Fig. 1 shows that trans-D2h appears at 113 K and cis-D2h at 295 K. The sample was prepared at room temperature. Where are the cis-D2h conformers at 113 K?

(2) The position of Fig. 1c is strange and can be misleading. The reviewer suggests to enlarge it and put it at the right side of Fig. 1.

(3) On page 10, "Notably, the "open-chain" organometallic bonding modes of the cis,cis tpy terminals have not been found in the experiments, indicating that it is the stable linear tpy-Fe₃-tpy coordination motif that leads to the chemical welding of the cyclic C-Au-C organometallic products." The meaning of this sentence is unclear. What does "open-chain organometallic bonding modes" mean?

(4) "II/III-modes to the SPy units holding Fe atoms with different numbers or configurations (c.f. SI Fig. S3)". It shall be briefly described in the main text what these filled pores are.

(5) What structures do m-DBTB form without adding Fe? Include such data in SI.

(6) In conclusion, "(i) the Fe-tpy coordination converts all the molecules to the cis-D2h conformers". This is not consistent with "Most of molecules (>90%) appear in the cis-D2h conformation" on page 8

(7) The Fe trimer with a Au atom is exotic though it is possible. Can the authors exclude other possibilities?

(8) The manuscript shall be improved by a native English speaker.

Dear editor,

We thank the reviewers for their comments and suggestions. We have re-analyzed our STM data, performed additional XPS experiments and extra DFT calculations to address the reviewers questions. The XPS results and more DFT calculations are included in the revised supplementary information (SI). We believe that these new data improve the quality of the manuscript to meet the criteria of the publication with Nature Communications. Our answers and revisions point-by-point are as follows.

(Other minor revisions include: 1, the author list is updated, including three coauthors who contribute the paper with XPS experiments; 2, Author contributions are modified accordingly; 3, several references are updated.)

Reviewer #1 (Remarks to the Author):

This manuscript describes the fabrication of N-doped porous carbon nanoribbons. The authors have designed control experiments to highlight the template effect of Fe-tpy coordination, which can control the on-surface polymerization to form linear nanoribbons. Each section is discussed in detail and shows a complete process. However, analogical on-surface polymerization with metal-coordinated template has been intensively reported by Lin's group (J. Am. Chem. Soc. 2013, 135, 3576, etc.). Not long ago, Aitor Mugarza et al. also reported bottom-up synthesis of nanoporous graphene by on-surface aryl-aryl coupling reaction and its electronic properties (Science, 2018, 360, 199). In terms of the criteria of publication with Nature communications, this manuscript lacks of deep insight and novelty, hence I could not recommend its publication. Detailed comments are shown below.

Answer: We thank the reviewer for her/his reviewing and comments. However, we respectfully disagree with the referee's comments that our work lacks novelty and is analogous to previous reports. The point of our work is the ability to fully control the on-surface reaction pathway, which represents a very important leap in the field of on-surface synthesis. This view is also shared by Reviewers #2 and #3. For instance, Reviewer #2 commented "This manuscript is an excellent example for a strategy that used reversible bonds for structure formation and induces the formation of (here irreversible) covalent bonds in a successive step." And Reviewers #3 commented "The data are in good quality and very convincing. Template effects are well known in organic synthesis. To my view, this work is the first work of successfully utilizing coordination as template to control on-surface reactions. It is an exciting breakthrough in the rapidly developed field of on-surface synthesis."

We note that both the metal-organic templated on-surface polymerization and nanoporous graphenes have been reported, as we also cited their work in the old manuscript. Lin's work (J. Am. Chem. Soc. 2013, 135, 3576, etc.) used template to control the size of the polymers but not to produce new products. Mugarza's work (Science, 2018, 360, 199) did not use template. Our work, however, is distinct from

them and other previous reports in this field, as we have successfully controlled the on-surface reaction pathway by using the non-covalent (metal-organic coordination) template, demonstrating a selective covalent coupling and successfully synthesizing a novel N-doped porous nanoribbon structure. Controlling the reaction pathway of on-surface synthesis is crucial but challenging towards the atomically precise fabrication of conjugated nanostructures, which has not been well addressed in previous reports, including the works the reviewer mentioned. Our approach demonstrates a new route for atomically precise synthesis of conjugated nanostructures on surfaces. By thoroughly revising the manuscript according to the reviewers' comments, we thus are confident that the revised manuscript meets the criteria of publication with Nature Communications.

1) In Figs.1a and b, the STM image of the Kagome lattice show that the "hexagonal pores" depict either dark or bright central protrusions. What is the difference between them?

Answer: We attribute the bright central protrusions to the confined molecules inside the pores. The confined molecules were rotating or vibrating thermally during scanning at 113 K, and thus showed a blur appearance. This phenomenon has been widely observed on other porous networks and been well understood [Palma, C.-A. *et al.*, *Nano Lett.* **14**, 4461-4468 (2014); Shi, Z. & Lin, N. *J. Am. Chem. Soc.* **132**, 10756-10761 (2010)]. The dark pores are the pores without confined molecules inside.

Revision: We add a sentence to explain this phenomenon. See line 19-21, page 5.

"The donut-shaped species observed in the lattice voids could be assigned as the non-bonding *p*-DBTB monomers, which presented blur appearance due to the thermal rotation and vibration⁴⁹."

2) Line 18, page 1, the expression "debromination aryl-aryl coupling reaction" seems weird and should be replaced by "debrominated" or "aryl-aryl coupling reaction after debromination". Such an expression is used frequently in the text.

Answer: We thank very much the reviewer for the suggestion.

Revision: We have modified the expression, by using "debrominated". The expression "dehalogenation" was also replaced by "dehalogenated". See line 23, page 1; line 9, 12, 33, page 2; line 14, 16, 17, page 6; and etc.

3) Line 21, page 1, "yielding the pure phase of SPy-p-Ph porous nanoribbons" is skeptical because according to the text, the ultimate products based on SPy-p-Ph have three modes.

Answer: Thanks for the reviewer to point this out. We agree this sentence might be misleading. We intend to highlight that our strategy can yield fully conjugated SPy-p-Ph porous nanoribbons instead of the inhomogeneous polymorphic structures (in the absence of Fe) as illustrated in Figure 2. The three modes are assigned so according to how the SPy pores hold (or not) the Fe atoms. Thus, these modes are not about the formation of the organic SPy-p-Ph nanoribbons.

Revision: We revise the sentence, to make the description precise. See the text in line 26-27, page 1.

“..., and finally yields the pure organic sexipyridine-*p*-phenylene (SPy-*p*-Ph) nanoribbons.”

4) Line 25, page 4, “The high-resolution STM image provided in Fig. 1b displays that three molecular monomers tilted by 60° are linked together via Br...Br interaction to form a trimeric cluster.” The Br...Br interaction as well as the formed trimers should be marked in Fig. 1b.

Answer: We thank the reviewer’s suggestion.

Revision: We have modified Fig. 1b to highlight the trimers, and provided an additional figure (Fig. 1c) to manifest the Br...Br interaction. See the modified Figure. 1 in page 4.

5) Line 10, page 5, “The theoretically estimated torsional barrier for the *trans/cis* transition of a 2,2'-bpy molecule in free space is about 34 kJ/mol, much higher than that for the *cis/trans* transition ~6 kJ/mol⁴⁹.” This sentence is in contradiction to the reference whose Fig. 2 actually includes the thermodynamic information which supports that the *trans* configuration is more stable.

Answer: The sentence describes and compares the torsional barriers of *trans*-to-*cis* and *cis*-to-*trans* transition of 2,2'-bpy molecule, to support that the *trans* conformer is more stable (because of a higher barrier for the *trans*-to-*cis* transition). This indeed agrees with the Fig.2 in the reference. We revised the sentence to avoid misleading.

Revision: We revise the sentences, pointing out the *trans* configuration is a favorable state. See line 22-24, 25-26, page 5.

“As suggested by the theoretic calculations⁵⁰, the energy of a *trans*-shaped 2,2'-bpy molecule is 34 kJ/mol lower than its *cis*-conformer, ...”; “Thus, the *trans*-D_{2h} conformer represents the most stable state among all the conformers.”

6) Line 12, Page 6, “The thermal annealing at 157 °C initiated...an inhomogeneous polymorphic structure (Fig. 2a). ” Meanwhile, the caption of Fig. 2a is “STM overview of the sample following an annealing treatment at 388 °C. ” What is the real temperature then?

Answer: We thank the reviewer for pointing out this issue. Both temperatures are real. The aryl-aryl coupling reaction occurred within a large temperature window. 157 °C was the lowest annealing temperature after which we observed the polymorphic covalent structure, while 388 °C was the highest annealing temperature we used, after which we observed the structure being still stable.

Revision: We have modified the text, by quoting “Fig. 2a” after the corresponding text, to avoid such misleading. See line 19-20, page 6.

7) Line 5, page 8, “The Fe was used for its capability to coordinate with the tpy endgroups, which has been well documented both in solution and on surfaces⁵⁰⁻⁵⁴.” But Ref. 52 only reports the assembly of the *cis*-BTP-TPE molecules with the tpy

group on Cu(111), and has nothing to do with Fe. Such a citation is redundant.

Answer and revision: We have deleted the citation here. See line 22, page 8.

8) Line 8, page 9, the digital “3” in “Fe₃ clusters” should have a subscript.

Answer and revision: We have modified the description of the model of the organometallic chains. And, in the revised manuscript, Fe₃ or Fe₃ related models are presented in SI, where we have used the subscript. See line 9, 19, 20, etc., page 6, in SI.

9) Line 3, page 10, “The image indicates two extra ‘spots’, each in the ‘bay’ of the tpy terminals.” This sentence is unclear because there are too many bright spots in Fig. 4c. The authors should clarify the two specific spots and mark them in the figure.

Answer and revision: We have modified Figure 4. Different symbols are used to mark the spots in the new figures; see Figure 4b,c and the caption, page 10. The descriptions in the main text are also made; see page line 23-24, 27-28, page 9; line 19-20, page 10; line 1-3, page 11.

“Three linearly aligned “spots” (marked by symbols + or x, Fig. 4b) are discernible ...”; “The small dots (o) discernible in the inter-chain spaces (Fig. 4b) ...”; “The topograph indicates that there is one additional spot (white “x”) located beside the central spot (black “x”) and in the bay of each tpy terminal.”

10) Line 2, page 15, the authors mention in the experimental section “with the sample temperature at 100 K–293 K”. The specific temperature should be indicated in each figure to judge the imaging quality and the surface states. In Line 3, “The STM images were processed by using the software WSxM” should provide the reference: Horcas, I. et al, WSXM: A software for scanning probe microscopy and a tool for nanotechnology. Rev. Sci. Instr. 78, 013705 (2007).

Answer and revision: We added the temperatures in the captions. We added the reference; see citation 63 (page 16) and the reference in line 42, page 19.

11) There are impurity molecules in the sample, making the image quality in Figs. 1 d and f poor.

Answer: We thank for the comment. I suppose that the “impurity molecules” the reviewer referred to is the “bright dots” in the close-packed island. They are not impurities, as also indicated in our low-temperature scanning in Fig. 1a and b. We propose that the “dots” are induced by Br atoms that spontaneously protrude upwards from the substrate. Because the images were acquired at room temperature, gas-phase molecules were thermally migrating, leading to a poor imaging quality. Nevertheless, the images are provided to show the conformational flexibility of the molecules, which is manifested by the conformational transition monitored exclusively by the real-time scanning at room temperature.

12) In Fig. 1f, the C_{2v} molecule marked by the white arrow in the left panel does not transform into the cis-D_{2h} isomer after image scanning, which should be pointed out

in the text.

Answer: We thank for the reviewer's suggestion.

Revision: We have modified the caption and the text to point this out. See the caption of revised figure 1g, and the text in line 5-6, page 6.

13) The caption for Figs. 2e-f is ambiguous for the statistical analyses of the molecular conformations and coupling modes.

Answer and revision: We have modified the Fig. 2b-d, where we mark the coupling modes by using ovals in different colors, while the conformations have been clearly defined in the context. The corresponding modifications in the captions and the text have also been made. See modified figures 2b-d, and the caption, in page 7. See modified text in line 2-5, page 5; line 28, 29 and 33, page 6; line 3, page 8.

14) The imaging temperature should be lowered to increase the STM image resolution. For instance, the coordinating Fe atoms could not be discerned in Fig.3.

Answer: We thank the reviewer's suggestion. Due to the instrumental conditions, the imaging temperature lower than 100 K is not accessible. After all, the cis-D2h conformers were clearly revealed, and distinct from the sample in the absence of Fe. This fact leads us to conclude the Fe-tpy coordination inducing the conformational transformation. Therefore, the images presented are sufficient to support the major conclusions.

Revision: We have cited the references 49 and 56 to reminder readers about the phenomenon that the coordinating Fe atoms might be invisible in STM topography. See line 5, page 9.

15) More data should be provided to evidence the formation of the Fe₃ clusters and the enantiomeric F atom positions in II/III mode enantiomers in Fig. 4.

Answer: We thank for the review's suggestion. We have re-analyzed our STM data, performed additional XPS experiments and more DFT calculations. These new data and analysis lead us to maintain the model proposed in the old manuscript, where the bright feature of the Fe-tpy coordination is suggested to be induced by the Au adatom beneath the central Fe atom. II/III modes were presented in Fig. 5.

Our discussion is listed briefly as follows:

i) By analyzing the STM data, we find the ratio of Br:molecule in the domains of the L-mode chains being 4. At the domains edges, the missing of Br atoms can be observed. The amount of these Br atoms should be too few to make all of the central Fe atoms "bright" (by binding to Fe), while our STM observation shows that all of the central Fe atoms (of the L-mode links) display the bright feature. This contradiction leads us to propose that the bright feature is not likely to be caused by the binding of Br atom(s).

ii) Our XPS measurements (using monochromatic synchrotron radiation) to the organometallic phase show Br 3d doublet peaks at 68.0/69.0 eV (Fig. A1.1), indicating only one Br species that can be assigned to the chemisorbed Br adatoms on the Au(111) surface, according to literatures [Basagni, A. *et al.*, *Chem. Comm.* **51**,

12593-6, (2015); Smykalla, L. *et al.*, *Nanoscale* **7**, 4234-4241, (2015); Batra, A. *et al.*, *Chem. Sci.* **5**, 4419-4423 (2014); Krasnikov, S. A. *et al.*, *Nano Res.* **4**, 376-384, (2011)]. In agreement with the STM data, the XPS result suggests that no Br-Fe binding exists on the organometallic phase.

Fig. A1.1 | XP spectra of the Br 3d, for the organometallic phase (upper) and the conjugated phase (lower). The two phases were obtained with the annealing treatment at 423 K and 603 K, respectively.

iii) Our previous DFT calculations have excluded the model of Fe₃ without Au adatom beneath, although the Fe₃-tpy coordination was demonstrated feasible on Ag(111) surfaces [Schiffrin, A. *et al.*, *ACS Nano* **12**, 6545-6553 (2018); Krull, C. *et al.*, *Nature Communications* **9**, 3211 (2018)]. Our additional DFT calculations considering one Fe atom beneath the central Fe (of the Fe₃ cluster; namely Fe₃-Fe model) suggest that the model is unstable, collapsing to an Fe₃ cluster with the underneath Fe atom diving into the substrate. Thus, we attribute the bright feature of the central Fe to the Au adatom underneath; the STM simulations reproduce well the major features with our STM observation. The incorporation of Au into the Fe-tpy coordination is likely to be associated with the aurophilicity of the Au atoms [Zhang, H. & Chi, L. *Adv. Mater.* **28**, 10492-10498 (2016)].

Last, although the attribution of Br-Fe binding can be excluded, the determination of the Fe-clusters structure based solely on theoretical calculations is somehow not satisfactory. However, unambiguously determining the state of the central Fe atom needs state-of-the-art techniques, including tip-manipulation and scanning tunneling spectroscopy working at cryogenic temperatures [Schiffrin, A. *et al.*, *ACS Nano* **12**, 6545-6553 (2018); Krull, C. *et al.*, *Nature Communications* **9**, 3211 (2018)], and thus cannot be done with our instrument.

Revision: We revise the description of the model to make the point clear; see line 4-8, page 11 in the main text.

We provide the XPS measurement results in SI, to support our model; see Figure S3, page 4-5 in SI.

We provide the additional DFT calculations to further exclude an Fe₃-Fe model; see Figure S4d, page 6-7 in SI.

16) In Fig 4a, the marked surface lattice orientation is wrong. It should be [1-10] rather than [110] because the latter does not exist on Au(111).

Answer and revision: We have corrected it in the revised manuscript. See Fig. 4a, page 10.

17) Some molecules in Fig.S1 seem to have already formed the structures similar to those Fig.4, which is different from description in the text.

Answer: I suppose that the reviewer refers to “some molecules” as the short chains in Fig. S1 that do resemble those in Fig. 4. However, the short chains were solely induced and temporarily stabilized by coordination between Fe (clusters) and tpy terminals. These chains and other molecular clusters were much unstable, and coexisted with the gas-phase molecules, which were invisible in the STM images acquired at room temperature. The similar sample acquired at low temperature (Fig. 3) revealed that most of molecules were intact (as indicated by their apparently discernible cis-D_{2h} conformations). All these features make the structures in Fig. S1 different from the stable C-Au-C organometallic chains shown in Fig. 4.

Revision: We revised the text to make the point clear. See line 11-13, page 9.

“Note that these assembled chains became unstable and were partially dissociated at room temperature, and only short chains or small molecular clusters were resolved occasionally (Fig. S2).”

18) In Fig. 5d, the bias of high-resolution STM images is -1.5 V which is different from that in Fig.5b (STM overview, -0.5V). Please explain why not choose the same bias?

Answer: We did not show the tunneling condition of Fig. 5b in the old manuscript. The reviewer might refer to Fig. 5e, which does be acquired with the bias of -0.5 V, different from that for Fig. 5d. We have conducted many experiments using different tunneling conditions, confirming an independence of the molecular topographic feature on the bias voltages. The images of Fig. 5d and 5e are selected out because each image is presented with the best resolution.

Revision: We added the information on the acquisition conditions of all STM data, to help readers to evaluate the image quality; for instance, see line 17, page 12.

19) In Figs. 4 and 5, the structural analyses involving the attribution of the bright spots between the chains or inside the cycles are not convincing merely by the calculated simulations. Such an attribution is rather weak: i) How to attribute the central spots for the L-mode link in Fig. 4c and the bright spots inside the Spy in Figs.

5d and 5e to the Fe or Au atoms? ii) To explain the big central bright spot in the L-mode link, the authors add an Au atom beneath the central Fe atom in their calculation (Fig. 4d) which does not agree with experimental results in Fig. 4c.

Answer: This question is about the detailed structure of the L-mode link in Fig. 4 and the II/III-mode in Fig. 5, similar to Question (15). After re-analyzing our STM data, performing additional XPS experiments and extra DFT calculations, we could confirm the structural models proposed in the old manuscript.

The complementary XPS measurements are performed to examine the reaction pathways we proposed. The Br3d signals disappeared at 603 K at which conjugated SPy-p-Ph ribbons formed. (See Fig. A1.1). This temperature is slightly higher (by 30 degrees) than the range of Br-desorption on Au(111) (~200-300 °C) reported in literatures [Krasnikov, S. A. *et al.*, *Nano Res.* **4**, 376-384 (2011); Batra, A. *et al.*, *Chemical Science* **5**, 4419-4423 (2014); Smykalla, L. *et al.*, *Nanoscale* **7**, 4234-4241 (2015)]. And the temperature is in the lower end of the annealing temperatures used in our STM trials, where the SPy-p-Ph ribbons emerged after annealing at 578-705 K. Thus, we propose that the II/III modes do not include Br atoms. Considering that both Fe and Au exist on the surface, and can be captured by py ligands, we proposed the models for the II/III mode SPy units, in which the Fe or Fe-related clusters are ascribed to the bright features of the modes. The DFT simulated topographs support the models.

Last, experimental determination of the bright feature of the L-mode links or the details of the II/III-mode SPy units perhaps needs state-of-the-art tip-manipulation or scanning tunneling spectroscopy working at cryogenic conditions [Schiffrin, A. *et al.*, *ACS Nano* **12**, 6545-6553 (2018); Krull, C. *et al.*, *Nature Communications* **9**, 3211 (2018)], and thus cannot be done using our instrument.

Revision: See line 8-10, page 13 in the main text.

The XPS results are provided in Fig. S3, page 4-5 in SI.

The DFT simulations are shown in Fig. S6, page 6 in SI.

20) In Fig 6g, the products are not in agreement with the modelled structures in the right panel.

Answer: We only considered the organic structures in the model, which was also the focus of this paper – the controlled covalent coupling. Thus, the non-covalent metal-coordination was not considered in the model.

21) The measuring distance should be unified, i.e., the center-to-center distance between neighboring molecules.

Answer and revision: We thank the reviewer for the suggestion. We have carefully read the manuscript and corrected all the descriptions on the measuring distances. For instance, see line 1, page 7; line 3 and 21-22, page 9, etc.

22) The unit of the temperature should be unified by using either K or °C.

Answer and revision: We use K to unify all of the temperatures in the manuscript.

23) It's suggested that the co-existing products in the structures over the transition temperature ranges should be provided to support the proposed reaction pathways.

Answer: We thank for the reviewer's suggestion. Such information has been shown in Figure 6a, where coexistence of the C-Au-C L-mode links with the C-C coupled III-mode SPy units for the m-DBTB molecules was visible. To make the whole paper concise and focus, we have not provided such information for the p-DBTB molecules in the revised manuscript.

Revision: We have described that the coexistence of L-mode with III-mode was visible in Figure 6a in main text. See line 25-27, page 13.

"Notably, a large number of the III-mode SPys have been observed at this annealing temperature relative to the other two modes of SPys, which is in agreement with the tendency revealed in Fig. 5f."

24) The authors should explain in detail why the introduction of the Fe atoms could change the reaction pathways to form the organometallic C-Au-C intermediates.

Answer: We thank the reviewer very much for pointing out this important issue. The C-Au-C intermediates have been reported in some literatures [Zhang, H. *et al.*, *J. Am. Chem. Soc.* **137**, 4022-4025 (2015); Urgel, J. I. *et al.*, *J. Am. Chem. Soc.* **139**, 11658-11661 (2017)], where the C-Au-C bonds were observed generally because the next C-C coupling step was inhibited by the great steric hindrance between the adjacent reactive organic units. In our case, the formation of the stable Fe-coordination cluster leads to a relatively large separation between two molecules. Such that the C-Au-C intermediate can be exclusively generated, because leaving a larger void than the cyclic C-C coupled product does.

Revision: We have added a paragraph to explain in detail the formation of C-Au-C intermediates. See line 9-17, page 11.

The two references 30, 58 are cited: Zhang, H. *et al.*, *J. Am. Chem. Soc.* **137**, 4022-4025 (2015); Urgel, J. I. *et al.*, *J. Am. Chem. Soc.* **139**, 11658-11661 (2017).

25) The band gap of GNRs makes it an attractive semiconductor material for its quantum confinement. The authors may explore in detail the electronic properties of the SPy-p-Ph, which would definitely make this work more striking.

Answer: We thank the reviewer for the suggestion. We agree that the electronic properties of the SPy-p-Ph ribbon would be an interesting topic. Thus, we have conducted additional DFT calculations to predict the band structure of the SPy-p-Ph ribbons, and presented the results in the revised manuscript. Experimentally characterizing the electronic properties of the ribbons, which perhaps include the influence of the substrate, is out scope of this paper, and will be done in future.

Revision: We have added a paragraph to describe the band structure of the ribbon; see line 19, page 12 to line 6, page 13.

The DFT predicted results of the band structure are provided in Fig. S5, page 8, in SI.

26) This text is full of typos and grammatical flaws, which makes the text hard to read.

Some examples are listed below, to just name a few.

Line 4, Page 2: "bulky chemistry" should be "bulk chemistry".

Line 17, Page 2: "coupling generally are..." should read as "coupling generally is...".

Line 17, Page 2: " has" should be " have".

Answer: We thank the reviewer for her/his reviewing. We have carefully corrected all the typos and grammatical flaws, and thoroughly revised the manuscript with the help of a native English speaker.

Revision: For instance, see line 8, 21, 23, page 2, etc.

Reviewer #2 (Remarks to the Author):

The manuscript under review reports on the on-surface Ullmann coupling on Au(111) assisted by a coordination template. Specifically, a precursor with two terpyridyl units and four Br is used. The presence of Fe atoms as a template-forming metal center leads to the formation of oligomer chains with regular N-heteromacrocyclic repeat units. In contrast, a disordered covalently bonded phase is obtained in the absence of the template metal. This manuscript is an excellent example for a strategy that used reversible bonds for structure formation and induces the formation of (here irreversible) covalent bonds in a successive step. By using a second precursor with a different geometry, the authors confirm that their approach is not limited to one successful example. While the formation of the final covalent products is well confirmed by the STM images, the discussion of the intermediates raised some serious questions, that can easily be addressed by additional measurements:

Answer: We thank the reviewer for her/his comments and suggestions. We have re-analyzed our STM data, conducted additional XPS tests and extra DFT calculations to address these questions. We believe that these new data improve the quality of the manuscript to meet the criteria of the publication.

1. Fe-tpy templated molecular phase: Figure 3 shows STM images of the presumably intact p-DBTB precursor with Fe on Au(111). It is stated that the images were acquired with the sample held at 100 K. However, I could not find any information about the highest temperature to which the sample was exposed. This is important, because Fe is likely able to dissociate the C-Br bond already at a lower temperature than the gold surface does. This point is especially crucial to explain the bonding situation in the chains in Figure 3 (see also next comment). XPS should be used to clarify whether the C-Br bonds are still intact or not.

Answer: We thank the reviewer for pointing out this important issue. The highest temperature is 293 K (room temperature). And we also confirm that the p-DBTB precursors in Fig. 3 are intact. The literatures [Mao, X. F. *et al.*, *Phys. Chem. Chem. Phys.* **15**, 12447-12450 (2013); Zhang, R. *et al.*, *Chem. Commun.* **53**, 1731-1734 (2017).] reported that the annealing treatments at 80, 150 or 180 °C, to the Au(111) surfaces predeposited with a Br-derivative porphyrin compound and Fe atoms, did not dissociate the C-Br bonds. Thereby, although we agree that Fe is likely able to

dissociate the C-Br bond at a lower temperature than the gold surface solely does, we propose that in our case the temperature for the C-Br bond dissociation should not be as low as the room temperature. Because of limited time, our XPS measurements have been performed to examine the organometallic intermediate phase and the final pure organic ribbon phase with corresponding annealing treatments. Thus, the critical temperature (of the C-Br bond dissociation) has not been addressed. Nevertheless, with the above information and the high-resolution STM images, we may ascribe the molecules in Fig. 3 to intact *cis-D_{2h}* conformers.

Revision: We have included the description on the preparation of this sample. Please see line 25-27, page 8.

“Following the codeposition of the *p*-DBTB molecule with Fe atoms on a Au(111) substrate held at 293 K, the sample was cooled down to 100 K for STM measurements.”

2. To form the proposed halogen bonds in Figure 3c, the Br atoms (or, rather, the C-Br bonds) have an unfavorable angle. Normally, angles between 120° and 90° are observed, whereas in Figure 3c the this angle is in the range of 150°-160°. At some point, the interaction should become repulsive because of the predominant interaction of the sigma-holes. Are there examples for such large angles in the literature?

Answer: Thanks for the reviewer’s comments. In our answer to Question (1), the molecules in the chain structure in Fig. 3c has been confirmed being intact. There do exist the large angles ~150°-160° in the model; the examples are also reported in literatures[Pham, T. A. *et al.*, *Chem. Commun.* **50**, 14089-14092 (2014); Bui, T. T. T. *et al.*, *Angew. Chem., Int. Ed.* **48**, 3838-3841 (2009)]. However, our following analysis on the details suggest that a net attractive interaction is feasible.

In the literatures, the halogen interactions are classified into two types, according to the angles of a CX...X bond. After re-analyzing our STM data and the structural model (Fig. A2.1), we find that in the structure both Br1...Br2 and Br3...Br4 are attractive type-II bonds, though Br2...Br3 is a repulsive type-I bond due to its symmetrical configuration with angles of 156° (the large angles the reviewer referred to). Thus, a net attractive interaction is feasible. To make the paper concise and focus, we discuss this in SI.

Fig. A2.1 | Close-up look of the structural model and the derived distances and angles of the Br-bonding motifs between two molecular units.

Revision: We add the details of the Br-Br interactions in SI. The Figure A2.1 provided

above is also attached as Fig. S1 in the discussion. See line 9-11, page 9, in the main text; Fig. S1, page 2, in SI.

3. Fe-tpy templated organometallic phase: The postulated bonding motif with 3 Au adatoms and 3 Fe atoms is very unusual, especially as the central Fe atom is very much elevated. There should be other, more convincing explanations. Could the bright features in Figure 4b represent iron bromide clusters? Considering that bulk iron bromide is much more stable than bulk gold bromide, one should expect that the Br adatoms bind preferentially to Fe. This question can be addressed by XPS investigations, which provide details about the chemical state of Fe and Br on the surface.

Answer: We thank very much for the reviewer's suggestion. We have re-analyzed our STM data, performed additional XPS experiments and DFT calculations. These new results lead us to maintain the model proposed in the old manuscript, where the bright feature of the Fe-tpy coordination is suggested to be induced by an extra Au adatom beneath the central Fe atom (of a trinuclear Fe cluster).

Our discussion is listed briefly as follows:

i) By analyzing the STM data, we find the ratio of Br:molecule in the domains of the L-mode chains being 4. At the domains edges, the missing of Br atoms can be observed. The amount of these Br atoms should be too few to make all of the central Fe atoms "bright" (by binding to Fe), while our STM observation shows that all of the central Fe atoms (of the L-mode links) display the bright feature. This contradiction leads us to propose that the bright feature is not likely to be caused by the binding of Br atom(s).

ii) Our XPS measurements (using monochromatic synchrotron radiation) to the organometallic phase show Br 3d doublet peaks at 68.0/69.0 eV (Fig. A2.2), indicating only one Br species that can be assigned to the chemisorbed Br adatoms on the Au(111) surface, according to literatures [Basagni, A. *et al.*, *Chem. Comm.* **51**, 12593-6, (2015); Smykalla, L. *et al.*, *Nanoscale* **7**, 4234-4241, (2015); Batra, A. *et al.*, *Chem. Sci.* **5**, 4419-4423 (2014); Krasnikov, S. A. *et al.*, *Nano Res.* **4**, 376-384, (2011)]. In agreement with the STM data, the XPS results suggest that no Br-Fe binding exists in the organometallic phase.

Fig. A2.2 | XP spectra of the Br 3d for the organometallic phase (upper) and the conjugated phase (lower). The two phases were acquired after the annealing treatments at 423 K and 603 K, respectively.

iii) Our previous DFT calculations have excluded the model of Fe₃ without Au adatom beneath, though the Fe₃-tpy coordination was demonstrated on Ag(111) surfaces [Schiffrin, A. *et al.*, *ACS Nano* **12**, 6545-6553 (2018); Krull, C. *et al.*, *Nature Communications* **9**, 3211 (2018)]. Our additional DFT calculations considering one Fe atom beneath the central Fe (of the Fe₃ cluster) suggest that the model is unstable, collapsing into the stable Fe₃ cluster. Thus, we attribute the bright feature of the central Fe that has an extra Au adatom underneath. The STM simulations reproduce well the major features with our STM observation. The incorporation of Au into the Fe-tpy coordination is likely to be associated with the aurophilicity of the Au atoms [Zhang, H. & Chi, L. *Adv. Mater.* **28**, 10492-10498 (2016)].

Last, although the attribution of Br-Fe binding can be excluded, we admit that the determination of the Fe-clusters structure based solely on theoretical calculations is somehow not satisfactory. However, unambiguously determining the state of the central Fe atom probably needs state-of-the-art techniques, including tip-manipulation and scanning tunneling spectroscopy working at cryogenic temperatures [Schiffrin, A. *et al.*, *ACS Nano* **12**, 6545-6553 (2018); Krull, C. *et al.*, *Nature Communications* **9**, 3211 (2018)], and thus cannot be done with our instrument.

Revision: We revise the description of the structure and propose the tentative model for the L-mode links in main text; see line 4-8, page 11.

We present more details on how we theoretically determine the structural model in SI; see Fig. S4, page 6-7.

The XPS results are also provided in SI. See line 30-32, page 9 in the main text;

Figures S3 and S3, page 4-7 in SI.

4. Conjugated SPy-p-Ph nanoribbons: Some of the knots (pores) in the ribbons contain Fe centers, while others are empty. The authors explain this with a tendency of Fe to diffuse to the bulk and cite literature for metal-organic networks. However, there the presence of Br could change the situation here. Perhaps the Fe centers can be more easily released because they bind strongly to Fe? Are there any possibly FeBr_x related features visible in the STM images? Again, XPS could show whether there is Br on the surface after the annealing procedure or not. With the Fe on the surface, Br may desorb at much higher temperature than from "clean" (i.e., Fe free) gold.

Answer: Thanks for the reviewer's suggestion. We admit that the Fe atom may easily bind to Br atoms. But, we did not observe FeBr_x related features under our experimental conditions. We have also conducted XPS measurements to address this issue. The Br3d signals disappeared at 603 K at which conjugated SPy-p-Ph ribbons formed (Fig. A2.2). This temperature is 30 degrees higher than the range of Br-desorption on Au(111) (~200-300 °C) reported in literatures [Smykalla, L. *et al.*, *Nanoscale* **7**, 4234-4241 (2015); Batra, A. *et al.*, *Chemical Science* **5**, 4419-4423 (2014); Krasnikov, S. A. *et al.*, *Nano Res.* **4**, 376-384 (2011)]. Because of limited time, only two annealing temperatures were used, so that the critical temperature (for Br-desorption) cannot be determined or compared with that for the "clean" gold. The temperature 603 K is in the lower end of the annealing temperatures used in our STM trials, where the SPy-p-Ph ribbons emerged after annealing treatments at 578-705 K. Thus, we propose that the II/III modes do not include Br atoms, and their models base on DFT simulations are maintained in the revised manuscript.

After all, because of the clear tendency shown in our stepwise postannealing treatments (Fig. 5f), II/III modes can be ascribed to the species (Fe atoms or Fe-related clusters) non-covalently interacting with SPy hosts. The formation of these non-covalent species should not significantly affect the reaction process of the covalent SPy-p-Ph ribbons.

Revision: We present the XPS data to exclude the Br-Fe_x species in the conjugated ribbon phase; see Fig. S3, page 4-5, in SI.

We confirm the II/III modes being SPy units with Fe atoms or Fe-related clusters non-covalently interacted, on the basis of the results of the stepwise postannealing treatments; see line 8-15, page 13.

Reviewer #3 (Remarks to the Author):

In this manuscript, Xing et al report a study of using coordination template effects to achieve homogeneous holey nanoribbons. By comparing with the control experiments without using Fe, they demonstrated that the template play critical roles to control on-surface synthesis. Moreover, they used another precursor

molecule with similar terpy functions to prove this concept is generally applicable in on-surface synthesis. The data are in good quality and very convincing. Template effects are well known in organic synthesis. To my view, this work is the first work of successfully utilizing coordination as template to control on-surface reactions. It is an exciting breakthrough in the rapidly developed field of on-surface synthesis. I think this work will be highly appreciated in this community and thus strongly support its publication in Nature Communication after some minor revisions being made addressing the comments listed below.

Answer: We thank the reviewer for her/his reviewing and comments, especially for her/his recognition to our work.

Comments:

(1) Fig. 1 shows that trans-D2h appears at 113 K and cis-D2h at 295 K. The sample was prepared at room temperature. Where are the cis-D2h conformers at 113 K?

Answer: We thank the reviewer's comment. We propose that most of cis-D2h conformers at 295 K transformed into trans-D2h conformers when the sample cooled down to 113 K.

The island (in Fig. 1d) composed mainly of the cis-D2h conformers was coexisting with gas-phase molecules, which we have already described in the text. The islands were unstable and subject to dissolve during scanning at room temperature. Thereby, we did not count the cis-conformers or compare different conformers at different temperatures. However, in the island we were able to detect spontaneous conformational transition of the molecular monomers activated by thermal energy. With the observations, we manifested the excellent flexibility of the molecules, which was one of the prerequisites for us to control the reaction pathway by using Fe-coordination. Since the trans-D2h conformers represent a thermodynamic favorable state (as we have demonstrated in the main text), the cis-D2h conformers at 295 K are proposed to transform into trans-D2h conformers at 113 K.

Revision: We have revised the text to clarify that the islands coexisted with a large amount of gas-phase molecules. See line 8-10, page 6.

"Note that the chain-like structures were loosely packed and subject to dissolve during the scanning. Thus, there existed a large amount of gas-phase molecules, whose conformations were unknown."

(2) The position of Fig. 1c is strange and can be misleading. The reviewer suggests to enlarge it and put it at the right side of Fig. 1.

Answer and revision: We thank the reviewer's suggestion. We have modified and rearranged the figures in Fig. 1; see the modified Fig. 1, in page 4.

(3) On page 10, "Notably, the "open-chain" organometallic bonding modes of the cis,cis tpy terminals have not been found in the experiments, indicating that it is the

stable linear tpy-Fe₃-tpy coordination motif that leads to the chemical welding of the cyclic C-Au-C organometallic products.” The meaning of this sentence is unclear. What does “open-chain organometallic bonding modes” mean?

Answer: We thank the reviewer for pointing out this issue. The “open chain organometallic bonding mode” means that, for instance, each cis,cis tpy terminal binds with two neighboring cis,cis tpy terminals through one of its py radicals, thus forming an open-chain structure. This bonding mode is acyclic, in contrast with the cyclic organometallic coupled L-mode links.

Revision: We have chosen the word “acyclic” to describe this bonding mode; see line 17-18, page 11.

(4)“II/III-modes to the SPy units holding Fe atoms with different numbers or configurations (c.f. SI Fig. S3)”. It shall be briefly described in the main text what these filled pores are.

Answer and revision: We have modified the description in the main text; see line 8-10, page 13. “the SPy units (I-mode) could accommodate Fe atoms (II-mode), or Fe-related clusters (III-mode), and thus exhibit different contrasts on the central parts; see SI, Fig. S6 for the theoretical simulations.”

The additional discussion on the modes is provided in Supporting Information; see Fig. S6, page 9, in SI.

(5)What structures do m-DBTB form without adding Fe? Include such data in SI.

Answer and revision: We thank the reviewer’s suggestion. The results show inhomogeneous polymorphic structures, including diverse coupling modes. We have included the data in SI; see Figure S7, page 10, in SI.

(6)In conclusion, “(i) the Fe-tpy coordination converts all the molecules to the cis-D_{2h} conformers”. This is not consistent with “Most of molecules (>90%) appear in the cis-D_{2h} conformation” on page 8

Answer and revision: We have revised this sentence to avoid misleading; see line 3-4, page 15. “The Fe-tpy coordination converts most of molecular precursors into cis-D_{2h} conformers, ...”

(7)The Fe trimer with a Au atom is exotic though it is possible. Can the authors exclude other possibilities?

Answer: We thank the reviewer’s comment. Experimentally determining the detailed structures of the Fe cluster perhaps needs state-of-the-art techniques, such as tip-manipulation or scanning tunneling spectroscopy working at cryogenic conditions [Schiffrin, A. *et al.*, *ACS Nano* **12**, 6545-6553 (2018); Krull, C. *et al.*, *Nature Communications* **9**, 3211 (2018).], and thus cannot be done using our instruments. Alternatively, we have re-analyzed our STM data, performed additional XPS experiments and extra DFT calculations to support our proposed model. On the basis of these new results, it is suggested that the model we have presented is feasible, while other configurations can be excluded.

The discussion is listed briefly as follows:

i) We have re-analyzed our STM data, which reveal that in the organometallic chain phase, the ratio of Br/molecule is 4. This excludes the scenario that the binding of Br atom(s) with the central Fe atom (within an Fe₃ cluster) causes the bright feature.

ii) Our XPS measurements (using monochromatic synchrotron radiation) to the organometallic phase show Br 3d doublet peaks at 68.0/69.0 eV (Fig. A3.1), indicating only one Br species that can be assigned to the chemisorbed Br adatoms on the Au(111) surface, according to literatures [Basagni, A. *et al.*, *Chem. Comm.* **51**, 12593-6, (2015); Smykalla, L. *et al.*, *Nanoscale* **7**, 4234-4241, (2015); Batra, A. *et al.*, *Chem. Sci.* **5**, 4419-4423 (2014); Krasnikov, S. A. *et al.*, *Nano Res.* **4**, 376-384, (2011)]. In agreement with the STM data, the XPS results suggest that the bright feature is not likely to be caused by Br-Fe binding.

Fig. A3.1 | XP spectra of the Br 3d, for the organometallic phase (upper) and the conjugated phase (lower). The two phases were obtained with the annealing treatments at 423 K and 603 K, respectively.

iii) Our previous DFT calculations have excluded the model of Fe₃ without Au adatom beneath, although the Fe₃-tpy coordination was demonstrated feasible on Ag(111) surfaces [Schiffrin, A. *et al.*, *ACS Nano* **12**, 6545-6553 (2018); Krull, C. *et al.*, *Nature Communications* **9**, 3211 (2018)]. Our additional DFT calculations considering one Fe atom beneath the central Fe (of the Fe₃ cluster) suggest that the model is unstable. Thus, we attribute the bright feature of the central Fe to the Au adatom underneath; the STM simulations reproduce well the major features with our STM observation. The incorporation of Au into the Fe-tpy coordination is likely to be associated with the aurophilicity of the Au atoms [Zhang, H. & Chi, L. *Adv. Mater.* **28**, 10492-10498 (2016)].

We thereby propose the model of the L-mode link, in which the bright feature at

the center is attributed to the lifted central Fe atom because of the extra Au adatom beneath.

Revision: We have modified the description of the structural models; see line 1-8, page 11.

We have also cited the references [Schiffirin, A. *et al.*, *ACS Nano* **12**, 6545-6553 (2018); Krull, C. *et al.*, *Nature Communications* **9**, 3211 (2018)] in the revised manuscript.

To make the paper concise and focus, the discussion on the detailed structures of the L-mode links is included in SI; see the XPS results in Fig. S3, page 4-5 and DFT calculations in Fig. S4, page 6-7.

(8)The manuscript shall be improved by a native English speaker.

Answer and Revision: We have thoroughly and carefully revised the manuscript with the help of a native English speaker.

REVIEWERS' COMMENTS:

Reviewer #1 (Remarks to the Author):

While the revised version has largely addressed my previous concerns, I'm still not convinced that its novelty is satisfactory. In the authors' response, they state that "Our work, however, is distinct from them and other previous reports in this field, as we have successfully controlled the on-surface reaction pathway by using the non-covalent (metal-organic coordination) template, demonstrating a selective covalent coupling and successfully synthesizing a novel N-doped porous nanoribbon structure. Controlling the reaction pathway of on-surface synthesis is crucial but challenging towards the atomically precise fabrication of conjugated nanostructures, which has not been well addressed in previous reports, including the works the reviewer mentioned."

"Controlling the reaction pathway of on-surface synthesis is crucial but challenging towards the atomically precise fabrication of conjugated nanostructures, which has not been well addressed in previous reports..." is overstated. A piece of recent work (X. Zhou et al., "Steering Surface Reaction Dynamics with a Self-Assembly Strategy: Ullmann Coupling on Metal Surfaces", *Angew. Chem. Int. Ed.*, 2017, 56, 12852) mainly focuses on the controlling of the reaction pathways with metal-organic intermediates tweaked by the self-assembly strategy. Moreover, the metal-organic coordination can not be simply distinguished from the co-valent bonding described in this manuscript because herein the metal-C bond is very strong. Metal-organic coordination normally refers to the coordination of metal atoms to organic ligands via the approaching of the lone electrons of the ligands (i.e., associated with atoms like N, O, S, P, etc.) to the metal vacant orbitals (i.e., d orbitals). Therefore, conventional coordination definition is not suitable to describe the metal-C bond in this manuscript. Actually, surface science techniques have routinely demonstrated that the metal-C vibration is much higher in frequency than that between the atoms bounded by the weak metal orbital – lone electron coordination.

Reviewer #2 (Remarks to the Author):

The authors have addressed my comments with XPS measurements, comparison with literature data, and additional considerations. The related revisions of the manuscript fully resolve the open questions. Publication of this revised version in *Nature Communications* is recommended.

Reviewer #3 (Remarks to the Author):

The authors have performed additional XPS experiments and DFT calculations. These new results, together with the very careful revisions, make the paper much improved. All my previous comments are satisfactorily addressed. The comments of other two reviewers are also addressed fairly well.

The significance and quality of the revised manuscript definitely merit publication in *Nature Communication*. I believe this publication will make a long-term impact in the fast developing field of on-surface synthesis.

Response to referees

REVIEWERS' COMMENTS:

Reviewer #1 (Remarks to the Author):

While the revised version has largely addressed my previous concerns, I'm still not convinced that its novelty is satisfactory. In the authors' response, they state that "Our work, however, is distinct from them and other previous reports in this field, as we have successfully controlled the on-surface reaction pathway by using the non-covalent (metal-organic coordination) template, demonstrating a selective covalent coupling and successfully synthesizing a novel N-doped porous nanoribbon structure. Controlling the reaction pathway of on-surface synthesis is crucial but challenging towards the atomically precise fabrication of conjugated nanostructures, which has not been well addressed in previous reports, including the works the reviewer mentioned."

"Controlling the reaction pathway of on-surface synthesis is crucial but challenging towards the atomically precise fabrication of conjugated nanostructures, which has not been well addressed in previous reports..." is overstated. A piece of recent work (X. Zhou et al., "Steering Surface Reaction Dynamics with a Self-Assembly Strategy: Ullmann Coupling on Metal Surfaces", *Angew. Chem. Int. Ed.*, 2017, 56, 12852) mainly focuses on the controlling of the reaction pathways with metal-organic intermediates tweaked by the self-assembly strategy. Moreover, the metal-organic coordination can not be simply distinguished from the co-valent bonding described in this manuscript because herein the metal-C bond is very strong. Metal-organic coordination normally refers to the coordination of metal atoms to organic ligands via the approaching of the lone electrons of the ligands (i.e., associated with atoms like N, O, S, P, etc.) to the metal vacant orbitals (i.e., d orbitals).

Therefore, conventional coordination definition is not suitable to describe the metal-C bond in this manuscript. Actually, surface science techniques have routinely demonstrated that the metal-C vibration is much higher in frequency than that between the atoms bounded by the weak metal orbital – lone electron coordination.

Answer:

We thank the reviewer for her/his reviewing and comments.

The sentence overstated "..., which has not been well addressed in previous reports..." does not appear in the final version of the manuscript. Instead, a sentence "..., a comprehensive understanding of the template effect has not been fully developed." appears; see line 29-30, page 2.

We have also revised the main text, by deleting the overstated phrases, like 'new', 'novel' etc.; see line 32, page 2; line 1-2, page 14.

Indeed the report (X. Zhou et al., *Angew. Chem. Int. Ed.*, 2017, 56, 12852) has been included in the previous manuscript as Ref. 57, which is cited to compare with our coordination-template strategy; see "... reaction that takes place at higher temperature⁵⁷." in line 13-14, page 9. A similar paper (Chen, Q. *et al.*, *Angew. Chem. Int. Ed.*, 2017, 56, 5026-5030) describing the on-surface reactions guided by self-assembly has also been included as Ref. 41, which was cited as an example successfully using

supramolecular assembly strategy to control on-surface covalent reactions; see “...the reaction condition has to be well controlled³⁸⁻⁴¹.” in line 24, page 2.

We confirm that the metal-organic coordination template in our manuscript refers to the Fe-tpy coordination, that is, Fe-N coordination, which is a conventional coordination bond. The Au-C (metal-C bond) intermediate appearing in the whole manuscript has been expressed as organometallic/organogold bond explicitly.

To avoid doubt, we have modified a sentence to make the point clear; see “...the success of the highly self-recognizable, self-selective metal-organic coordination motifs (e.g., Fe/Cu-N/O bonds) in construction of ...” in line 26-27, page 2.

Reviewer #2 (Remarks to the Author):

The authors have addressed my comments with XPS measurements, comparison with literature data, and additional considerations. The related revisions of the manuscript fully resolve the open questions. Publication of this revised version in Nature Communications is recommended.

Answer: We thank for the reviewer’s comments and recommendation.

Reviewer #3 (Remarks to the Author):

The authors have performed additional XPS experiments and DFT calculations. These new results, together with the very careful revisions, make the paper much improved. All my previous comments are satisfactorily addressed. The comments of other two reviewers are also addressed fairly well.

The significance and quality of the revised manuscript definitely merit publication in Nature Communication. I believe this publication will make a long-term impact in the fast developing field of on-surface synthesis.

Answer: We thank for the reviewer’s comments and recommendation.